



# Reviews and Syntheses: Composition and Characteristics of Burrowing Animals along a Climate and Ecological Gradient, Chile

Übernickel, Kirstin[1], Pizarro-Araya, Jaime[2], Bhagavathula, Susila[1], Paulino, Leandro[3], Ehlers, Todd A.[1]

[1] Department of Geosciences, University of Tübingen, 72076 Tübingen, Germany
[2] Laboratorio de Entomología Ecológica, Departamento de Biología, Facultad de Ciencias, Universidad de La Serena, La Serena, Chile
[3] Departamento de Suelos y Recursos Naturales, Facultad de Agronomía, Universidad de Concepción, Chillán, Chile

*Correspondence to*: Kirstin Übernickel (kirstin@uebernickel.net)

**Abstract.** Although the burrowing activity of some species (e.g. gophers) is well studied, a comprehensive inventory of burrowing animals in adjacent biomes is not yet known, despite the potential importance of burrowing activity on the physical and chemical evolution of Earth's surface. In this study, we review the available information with a focus on: a) an inventory of burrowing vertebrates and invertebrates along the climate and ecological gradient in Chile; b) the dimensions and characteristics of burrows; and c) calculation of excavation rates by local species compositions. Methods used include a literature compilation (>1000 studies) of Chilean burrowing animal species integrated with global, species-specific excavation rates. A field study augments literature findings with quantification of the zoogeomorphic effects on hillslope mass transport at the animal community level and along the arid to humid-temperate climate gradient within the Chilean Coastal Cordillera (27-38° S latitude).

The literature review indicates 45 vertebrate and 345 invertebrate burrowing species distributed across Chile in different biomes. Burrowing depths for Chilean mammals range between 3 m (e.g. for skunks, *Conepatus*) to 0.25 m (for rock rats, *Aconaemys*). For invertebrates, burrowing depths in Chile range between 1 m for scorpions to 0.3 m for spiders. In comparison, globally documented maximum burrow depths reach up to more than 6 m for vertebrates (gopher tortoises and aardvarks) and 4 m for invertebrates (ants).

Minimum excavation rates of local animal communities observed from field sites in Chile are 0.34 $m^3\,ha^{-1}\,yr^{-1}$ for the arid site, 0.56 $m^3\,ha^{-1}\,yr^{-1}$ for the semi-arid site, 0.93 $m^3\,ha^{-1}\,yr^{-1}$ for the mediterranean site and 0.09 $m^3\,ha^{-1}\,yr^{-1}$ for the humid-temperate site, with the latter likely an underestimation. The calculated minimum Chilean excavation rates are within the large range of globally observed single species rates ranging between 0.01 and 146.20 $m^3\,ha^{-1}\,yr^{-1}$ for vertebrates and from 0.01 to 53.33 $m^3\,ha^{-1}\,yr^{-1}$ for invertebrates. Taken together, results highlight not only the diverse and latitudinally varying number of burrowing vertebrates and invertebrates present in different biomes, but also fosters the understanding of how burrowing activity changes over a gradient and is influenced by mean annual temperature, mean annual precipitation, slope aspect and latitudinal related incoming solar energy.

## 1 Introduction

Abiotic processes such as overland flow, creep, rain splash, wind and mass wasting, but also biotic processes induced by animals or plants influence the erosion of hillslopes (Amelung et al., 2018; Anderson and Anderson, 2010; Gabet et al., 2003; Smith and Gardner, 1985; Starke et al., 2020; Viles, 2020). Geomorphic processes that



result from burrowing animals are called "zoogeomorphologic" (Butler, 1995; Corenblit et al., 2011) and are a form of "bioturbation" (Hole, 1981; Wilske et al., 2015). Some of the earliest work documenting zoogeomorphologic processes (e.g. downhill sediment transport by animals) was Darwin's (1881) detailed description of earthworm activity. Burrowing animals in the pedosphere include earthworms; a wide range of insects (Bétard, 2020) including larval stages to adult forms of beetles, bees and ants; spiders and scorpions;

crustaceans; and myriapods; and also burrowing vertebrates such as lizards (e.g. Donoso-Barros, 1960) and larger mammals, like rodents, wolves, badgers (Thorp, 1949) and birds (e.g. Figueroa and Stucchi, 2008; Masello et al., 2006; Zavalaga and Alfaro-Shigueto, 2015).

However, despite the potential importance of bioturbation as a geomorphic process, inventories of burrowing vertebrates and invertebrates are limited and hamper our understanding of where and when they might impact

hillslope processes. Given this, here we investigate the hypotheses that burrowing animals composition and spatial occurrence varies in different biomes and the excavation rates of the animal communities will correlate with controlling factors. Current knowledge of burrowing activity is limited to very few species within specific biomes. Given this, uncertainty surrounds the importance of animal burrowing for geomorphologic studies. Zoogeomorphologic studies are not only infrequent, but also heterogenous in the methods applied (e.g. Butler et

al., 2013; Dietrich and Perron, 2006; Gabet et al., 2003; Platt et al., 2016; Wilkinson et al., 2009). As a result, it is difficult to determine surface downhill sediment transport rates at the ecosystem scale and on timescales relevant to surface processes studies (Dietrich and Perron, 2006; Platt et al., 2016). To determine these rates, an approach is needed to estimate the sum of all animal activity at a locality, and to understand the variation of the burrowing activity depending on the biome characteristics.

Previous studies of animal burrowing have focused on either a single species or, less frequently, on a few dominant species, with the vast majority of other animals remaining underrepresented. The most studied burrowing vertebrates are North-American gophers (e.g. Black and Montgomery, 1991; Dixon et al., 2009; Ellison, 1946; Huntly and Inouye, 1988; Kalisz and Stone, 1984; Seabloom et al., 2000; Smallwood and Morrison, 1999b; Thorn, 1978; Winchell et al., 2016), prairie dogs (e.g. Bangert and Slobodchikoff, 2006; Reading and Matchett, 1997),

zokors/mole rats (e.g. Zhang et al., 2003), or voles (e.g. Hall et al., 1999). The most studied invertebrate animals are ants, termites and earthworms (e.g. Bétard, 2020; Lobry de Bruyn and Conacher, 1990; Carlson and Whitford, 1991; Darwin, 1881; Lee and Foster, 1991; Viles et al., 2021; Wilkinson et al., 2009). Burrowing depth and excavation rates for more than one species per location have been documented in very few studies. Exceptions to this include work studying the interaction between species such as ants and small mammals (e.g. Eldridge and

Whitford, 2014) or communities with similar effects such as in rodents (Price, 1986). James et al. (2011) studied four vertebrate species in Australia and focused on their burrowing differences due to their native or non-native origin. Voslamber and Veen (1985) compared behavior of two mammal species (badgers & rabbits) in Belgium. In summary, downhill erosion and sedimentation by an excavating animal community at hillslopes or larger scales remain poorly understood.

Furthermore, existing studies do not cover a variety of biomes. There is a concentration of excavating animal studies in North American desert vertebrates and rodents (Platt et al., 2016). To our knowledge, only a single study compared the chemical effects of burrowing by comparable mammal species in different climate settings including a semi-desert in the Ural region and a spruce forest in the Moscow region (Abaturov, 1972). Studies conducted in different biomes and geographical areas, such as South America, are underrepresented in existing literature





(Haussmann, 2017). Thus, the effects of burrowing animals along gradients in temperature, precipitation, and vegetation are seldomly studied.

We build upon the previous global inventories and focus on burrowing vertebrates and invertebrates along the climate and ecological gradient in Chile. Our approach complements recent global compilation on ants as geomorphological agents in Viles et al. (2021) and a global review about insects as such agents (Bétard, 2020).

From existing literature, we compile the burrowing taxa of Chile, their distribution, burrow details such as tunnel size, depth and geographic extent, as well as excavation rates (when available). Globally published excavation rates of animals are summarized for comparison. In addition, we present new observations from four Chilean study areas spanning diverse biogeographic zones and document the number of burrow entrances of present taxa for vertebrates and invertebrates. From measurements of burrow entrances and tunnel lengths, taxa independent

excavation rates are calculated as minimum values. With this method it is possible to sample and estimate species independent zoogeomorphic effects at the hillslope scale, and along a climate gradient.

## 2 Methods

### 2.1 Literature compilation: Burrowing animals in Chile

To identify burrowing animal species in Chile and compile burrow related information, as well as global animal

excavation rates, a literature search was conducted using multiple online databases and published studies. Complementary searches were done in English and in Spanish. Additionally, available information was accessed at the scientific collections of Laboratorio de Entomología Ecológica de la Universidad de La Serena (LEULS), La Serena, Chile, and the División Aracnología del Museo Argentino de Ciencias Naturales "Bernardino Rivadavia" (MACN-Ar), Buenos Aires, Argentina. The species-specific information compiled included species

taxonomy, common name(s), alimentation type, special characteristics, group size, density, excavation characteristics (tunnel diameter, (maximum) burrow depths, details about burrow type), preferred habitat and distribution limit by elevation. For vertebrates, body size and weight were also compiled. The data compilation focused on species that excavate in an active manner. Excluded were species that are restricted to the use of burrows made by sympatric species, as well as ground dwellers that hide under rocks or vegetation and also soil-

dwelling animals that move through the soil by pushing material aside ("swimming") (Gabet et al., 2003). The full list of excavating species identified for Chile is provided in a companion data publication (Übernickel et al., 2020). Hereafter we use the data in this data publication for our figures and analysis. The subset of data regarding the four study sites were summarized as the burrowing species lists of vertebrates (Table S1) and invertebrates (Table S2). In addition, published excavation rates of animals worldwide were compiled (Table 1). For the calculation of

the excavation rates from volume measurements of publications, an intermediate near surface pedolith bulk density of 1.2 g cm$^{-3}$ was assumed (Amelung et al., 2018).

For visualization of mammals and invertebrates in Chile, their geographic range provided by the International Union for Conservation of Nature webpage (International Union for Conservation of Nature (IUCN) Red List of Threatened Species, accessed October 2018, n of downloaded species = 40), were merged and clipped in QGIS

(Version 2.18.26) ) to generate an overview map of the geographic extent in all of Chile. These results were compiled at a resolution of 0.5 latitudinal degrees (approx. 55 km) (Fig. 1). The topography, mean annual precipitation (MAP) and mean annual temperature (MAT) were also compiled (Fick and Hijmans, 2017). The distribution of invertebrate burrowing species in Chile are not yet available. To estimate the diversity of sympatric

species along Chile we generalized the mostly single study site information of the species to the extent of the

respective Chilean region (Übernickel et al., 2020) and plotted the amount of species per region (Fig. 2).

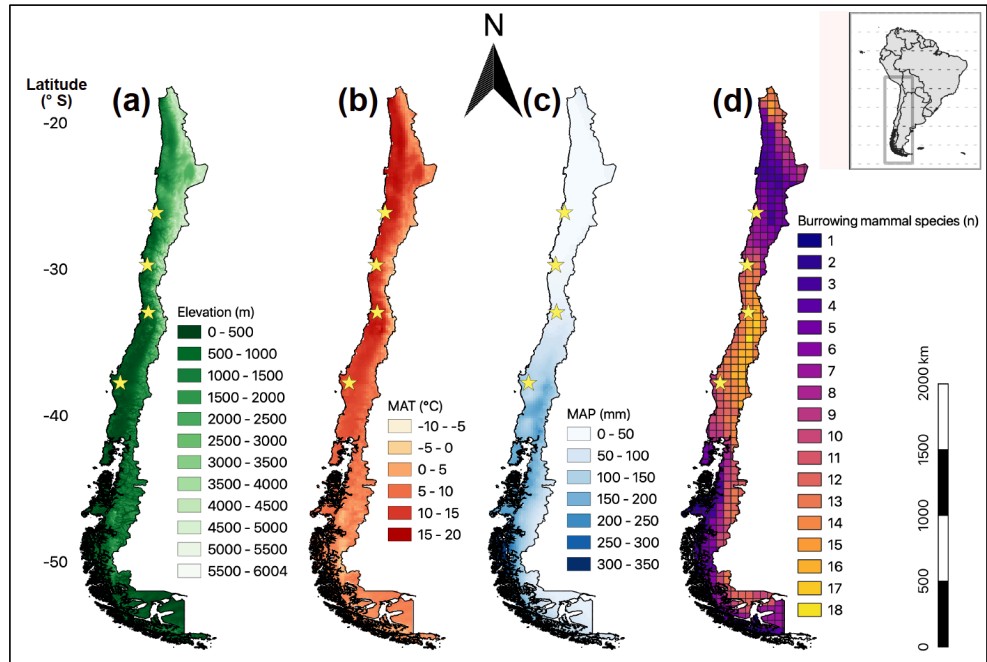

Figure 1: Map of mainland Chile showing four parameters. a: Elevation, b: Mean annual temperature (MAT), c: Mean annual precipitation (MAP), MAT & MAP from Fick and Hijmans (2017); d: Distribution of burrowing mammal species; in map d the grid size is 0.5 latitudinal degrees (approx. 55 km x 55 km side lengths), the data base for the depicted number of mammal
species are distributional polygons of 40 species (International Union for Conservation of Nature (IUCN) Red List of Threatened Species), downloaded from the webpage October 2018.

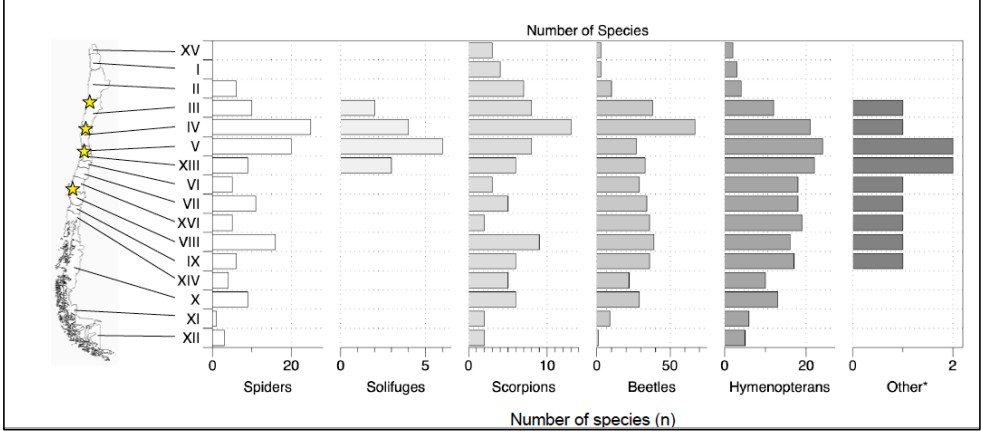

Figure 2: Distribution of burrowing invertebrates in Chile per Chilean region; The data compilation is based on the associated
data publication to this study, Übernickel et al. (2020). The map source is http://freevectormaps.com, Other* refers to Isopterans, Neuropterans and Hemipterans





## 2.2 Field study on hillslopes in Chile

### 2.2.1 Site description

The study sites are part of the German-Chilean priority program EarthShape (Earth surface shaping by biota; www.earthshape.net). The sites are located within three National Parks (NP) and one private reserve along the Chilean Coastal Cordillera to minimize anthropogenic disturbances. The latitudinal distance between these areas extends over 1,300 km and includes NP Pan de Azúcar (~26° S) in the arid north, the semi-arid Private Reserve Santa Gracia (~30° S), mediterranean climate NP La Campana (~33° S), and the humid-temperate NP Nahuelbuta

(~38° S) in the south (Fig. 3). For previous detailed information concerning site descriptions, soil analysis, as well as local vegetation see Bernhard et al. (2018), Oeser et al. (2018) and Schaller et al. (2018). In the following we summarize the climate, soil classification and topographic characteristics per location (Tabs. S1 and S2).

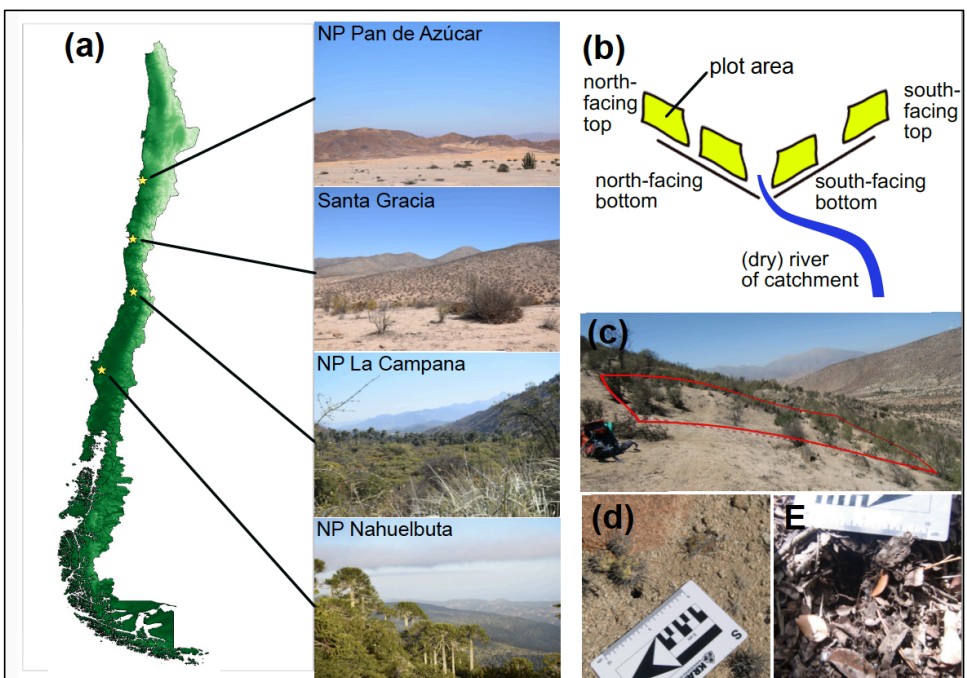

Figure 3: Overview of study sites in Chile. a: indication of study site locations (yellow stars) and example images; b: Plot setup
on opposing slopes with four plots per study site; c: Example image of plot in Santa Gracia, d-e: Examples of measured burrow entrances (D: on bare ground, Santa Gracia; E: within leaf litter, NP La Campana); Abbreviation: NP = National Park

NP Pan de Azúcar is the northernmost arid site with plot sites on hillslopes near -26.1098° S -70.5502° W. The NP was established in 1985 (Rundel et al., 1996). It has a MAP of 12 mm and a MAT of 16.8° C (Fick & Hijmans

2017). The elevation is 330 m a.s.l., the vegetation cover is < 10 % and the slopes range between 25° and 40° (Oeser et al. 2018), with the plots located on several slopes. The soil is classified as a Regosol. Bulk density varies between 1.3 Mg m$^{-3}$ at the north-facing slope and 1.5 Mg m$^{-3}$ at the south-facing slope and the texture is sandy loam in both slopes (Bernhard et al., 2018).

The Private Reserve Santa Gracia contains plots on hillslopes near 29.759° S 71.166° W. It has a MAP of 66 mm

and a MAT of 13.7° C (Fick & Hijmans 2017). The elevation is around 680 m a.s.l. and vegetation cover is 30-40 %. The north-facing slope has an angle of 15° and the south-facing slope of 25° (Oeser et al. 2018). The soil is



classified as Cambisol due to a differentiation between surface horizons and subsoil. Bulk densities are homogenous in the first 20 cm depth and 1.5 Mg m$^{-3}$ for both slopes, but texture predominates as sandy loam in the north-facing slope, while the south-facing slope has a loamy sand texture (Bernhard et al. 2018). Known

disturbances of this ecosystem are goats feeding on disturbed (reduced) vegetation (Armesto et al., 2007).

NP La Campana is the mediterranean site and contains plots at the sector Ocoa, around 32.9562° S 71.0637° W. The NP was established in 1967 with the boundaries legalized in 1985 (Macdonald et al., 1988). It has a MAP of 367 mm and a MAT of 14.1° C (Fick & Hijmans 2017). The elevation is around 730 m a.s.l., with a vegetation cover of close to 100 %, the north-facing slope has an angle of 12° and the south-facing slope of 23° (Oeser et al.

2018). The soil is classified as Cambisol. Bulk densities vary between 1.5 Mg m$^{-3}$ at the north-facing slope and 1.1 Mg m$^{-3}$ at the south-facing slope and the general soil texture is sandy loam for the first 20 cm of depth (Bernhard et al., 2018). Known disturbances of the ecosystem are (illegally) grazing cows (Rundel and Weisser, 1975).

NP Nahuelbuta is the southernmost site, with plots located near 37.8087° S 73.0137° W. This NP was established in 1936 (Servicio Agrícola y Ganadero, SAG, 1970). It has a MAP of 1,469 mm and a MAT of 6.6° C (Fick &

Hijmans 2017). The elevation is around 1,240 m a.s.l. and vegetation cover is ~100 %. The north-facing slope has an angle of 13° and the south-facing slope of 15° (Oeser et al. 2018). The soil is classified as an Umbrisol. The bulk density is 0.9 Mg m$^{-3}$ at the north-facing slope and 0.8 Mg m$^{-3}$ at the south-facing slope and the texture is sandy-clay loam (Bernhard et al., 2018).

### 2.2.2 Data compilation

Data acquisition took place during three field campaigns in November 2016, May 2017 and March 2018. Each site was visited during every field campaign. The plots per site were selected with the criteria to be located in east-to-west oriented valleys, two plots per opposing ~north and ~south facing slopes with one plot near both the top and one plot near the bottom of the hillslope (Fig. 3B). The size of the plots was 10 x 10 m in 2016 and 2018; in May 2017 5 m x 5 m sized plots were analyzed and numbers later scaled for comparability.

The burrow measurement procedure was to walk uphill within the plot in approximately 50 cm separated tracks, measuring the burrow entrances along the path and marking them to avoid duplicate measurements. The procedure and the parameters taken were optimized during the three field campaigns. First, the diameter of burrow entrances was digitally measured from photographs using ImageJ (v1.51m9). When the entrance was oval the major and minor axes were measured. In 2017 and 2018, the time effort was reduced by counting burrow entrances and taking

measurements on site. From 2017 onwards minimum tunnel length was determined as well, measuring manually from the entrance to the reachable maximum end of the tunnel. Furthermore, the direction of the tunnel into the ground was labelled as either vertical, horizontal or 45°. In 2018 the ground cover of the plots was estimated (Table S3). Note, for NP Pan de Azúcar the two plots per hillslope scheme was not completely fulfilled, as slopes at this site were too short. Plots were selected as similar to the original set up as possible.

### 2.2.3 Data analysis

From the collected data we correlated entrance diameter and minimum tunnel lengths (Fig. 4), measured the distribution of burrow entrances regarding the plot position on the top or bottom of the slope, distribution of entrances relative to the slope aspect (north- or south-facing), and distribution of entrances along the climate gradient (Fig. 5) and calculated minimum burrowing depth (Fig. 6) and minimum excavation rates (Fig. 7). Note,

the approach to evaluate the number of burrow entrances is a proxy to burrowing activity, as the actual density of




the individuals remains unknown. Also, by counting the burrow entrances, the number of burrows is not exactly registered, as one burrow may have several entrances. Statistical analysis on the burrow entrance data was not applicable as no replicates were collected. Finally, ground cover impeded the collection of sufficient data at the southernmost site (Nahuelbuta).

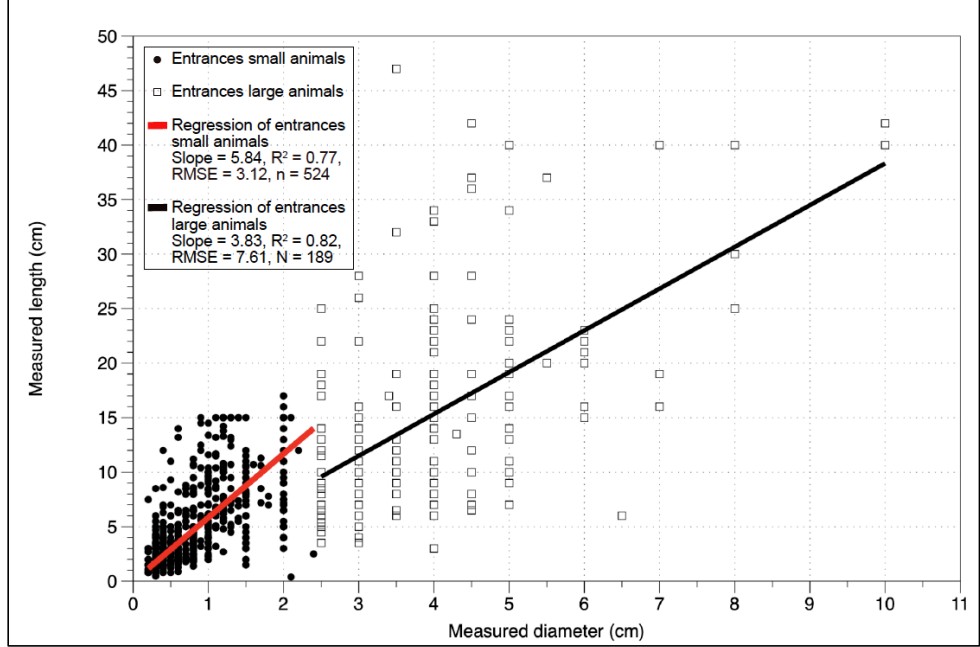


Figure 4: Linear regressions of the correlation between measured entrance diameter and tunnel lengths for the groups of large animals and small animals. The regressions are forced through 0 and are calculated from the data of 2018.

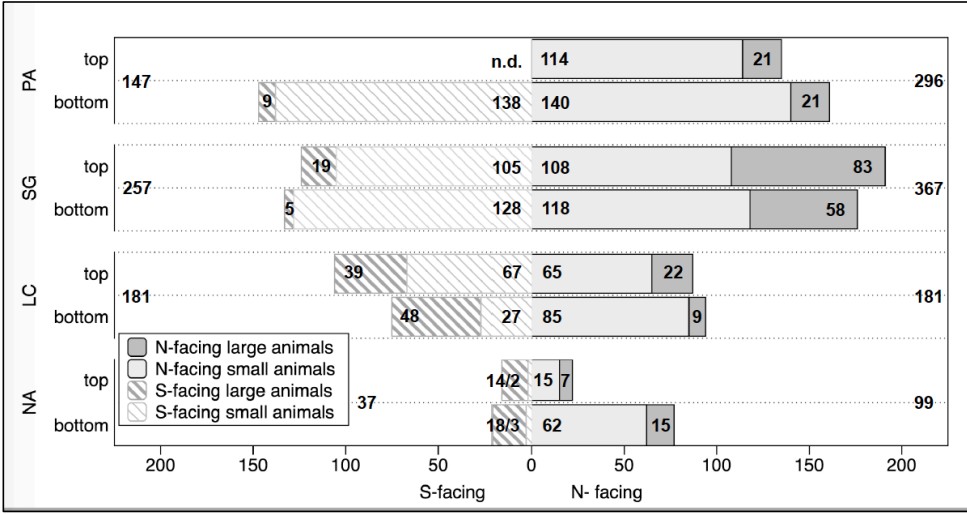


Figure 5: Overview of number of measured burrow entrances per plot specification across study sites. The data is presented per top/bottom position on slopes as well as N(orth)- or S(outh)- facing aspect. The numbers within the bars indicate the amount of entrances found at the respective slope position per animal group, the numbers outside of the bars are total values for the respective slope; the depicted data is pooled from all three field campaigns 2016, 2017 and 2018; Abbreviations: PA= Pan de Azúcar, SG = Santa Gracia, LC = La Campana, NA= Nahuelbuta, n.d. = not determined; *note that the spatial distribution of*



*the plots in NP Pan de Azúcar was not as uniform as at the other sites, the numbers contain three plots north-facing (1 top, 2 bottom) and one plot south-facing (bottom)*

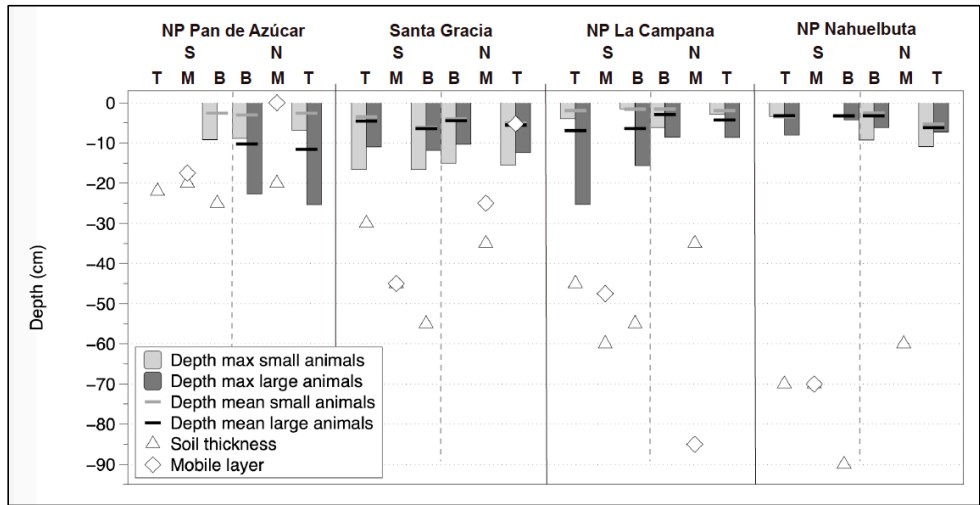

Figure 6: Animal burrowing depth as mean and maximum values from measurements in an arid to humid-temperate climatic gradient; for comparison are included: the depth of mobile layer following Schaller et al. (2018) and soil thickness following Oeser et al. (2018); abbreviations: PA= Pan de Azúcar, SG = Santa Gracia, LC = La Campana, NA= Nahuelbuta, NP = National Park, N = north-facing slope, S = south-facing slope, T-M-B are indications of the position on a slope: T= top-slope, M= mid-slope, B = bottom-slope

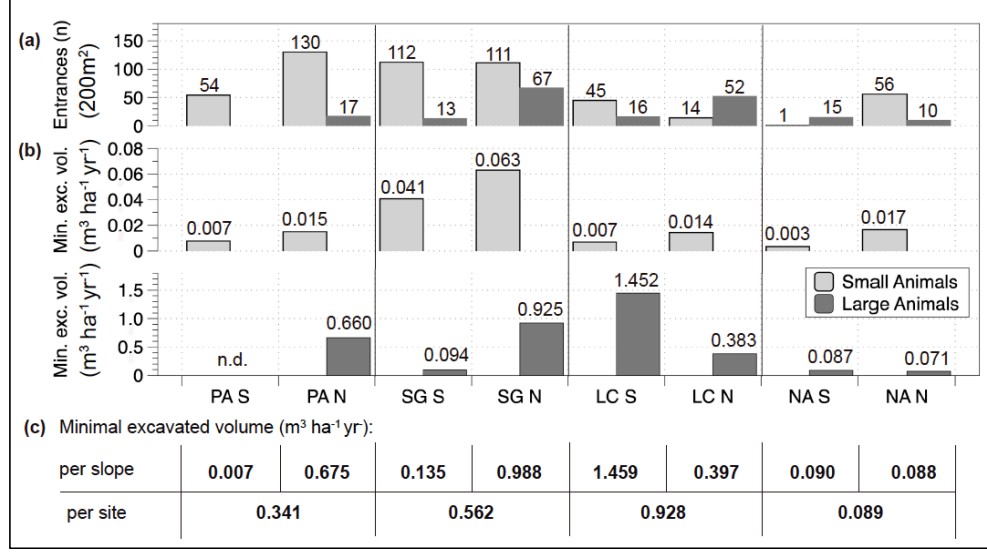

Figure 7: Burrow entrance numbers and minimum excavated volumes per slope in an arid to humid-temperate climatic gradient. a: Quantity of burrow entrances by animals separated by site, slope and animal group (large and small animals), n of entrances are sums of two plots per slope (composed of 2 10m*10m plots, total surface 10*20 m²); b: Minimum excavated volume of plots upscaled to m³ ha⁻¹ yr⁻¹ separated by slope; c: Minimum excavated volume amounts per slope and site summed for all animals; the figure is based on the data of the 2018 field campaign, Abbreviations: PA= Pan de Azúcar, SG = Santa Gracia, LC = La Campana, NA= Nahuelbuta, S = south-facing, N = north-facing, *note that in PA the sum of the south-facing plot entrances are of 1 plot, and the north-facing entrances sum of 3 plots.*

Minimum burrowed depths (Fig. 6) were approximated from the 2018 data, using the measured tunnel length, angle of the tunnel and the law of sines. The minimum excavation rate in m³ ha⁻¹ yr⁻¹ (Fig. 7) was estimated based

EGU Open Access

on tunnel volumes measured in March 2018. March is late summer in Chile, and represents the closest estimate of burrowing activity within a year before autumn and winter rainfall reset the surface. For each entrance the minimum excavated volume was calculated, dependent on the entrance's approximate shape. For round and oval entrances, the volume was calculated using the entrance radius, or major and minor axis, respectively. Crevice

shaped entrances were treated as ovals for conservative volume approximation. The volumes were converted from values per plot to $m^3\ ha^{-1}\ yr^{-1}$. Note that for tunnel volumes and excavation rate *minimum* values are reported, because additional volume behind a first curve or obstacle in the tunnel could not be measured.

In addition, the solar energy input and its effects may influence the distribution of the burrow entrances as they vary with the slope aspect (Gallardo-Cruz et al., 2009). To detect a link between of solar energy input and the

quantity of burrow entrances on north- and south-facing slopes, the diurnal course of potential direct radiation was calculated (Fig. 8). We assumed cloud free conditions and the potential direct radiation on an inclined surface using the terrain angle, based on the coordinates of the study sites following Bendix (2004). The calculations were conducted for the 21$^{st}$ June and December respectively, the period of highest and lowest solar radiation. For simplicity, atmospheric transmission is fixed to 0.8 and potentially occurring shadows of neighboring topography

and/or vegetation were neglected. Plotted values are for noon (12 p.m.).

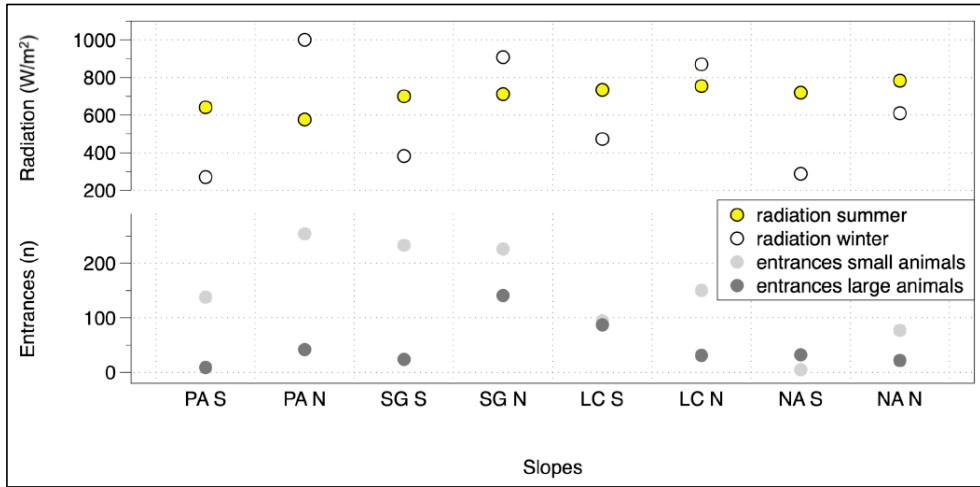

Figure 8: Calculated solar radiation and sum of animal burrow entrances separated per animal group (large and small animals) for the slopes along the studied climate gradient; the solar radiation calculation follows Bendix (2004); the slopes are oriented N(orth)-facing and S(outh)-facing; the depicted data is pooled from all three field campaigns 2016, 2017 and 2018;

Abbreviations: PA= Pan de Azúcar, SG= Santa Gracia, LC = La Campana, NA = Nahuelbuta, *note that in PA the sum of the south-facing plot entrances is of one plot, and the north-facing entrances sum of three plots.*

## 3 Results

### 3.1 Literature compilation on Chilean burrowing animal species

The literature survey included >1,000 references and revealed 45 vertebrate and 345 invertebrate species in Chile with excavating activity. The full species list is provided in the associated data publication (Übernickel et al., 2020). Of the 45 burrowing vertebrate species identified for Chile, 40 are mammals, two are birds and three are reptiles. Most burrowing species are rodents i.e. mice, (mole-) rats, rabbits, cavies (guinea pigs), chinchillas and nutrias. Other excavating mammals present include foxes, skunks, armadillos and some insectivores. Regarding



burrowing Chilean invertebrates, information beyond the name and taxonomic description from museum specimens was scarce, and many species remain unobserved in their habitat. Also, excavating activity in literature was frequently only indirect, e.g. only adaptations of body parts for burrowing activity were described. The 345 burrowing invertebrate species are composed as follows: within the basal group of diverse spiders, there are 83 species with 44 trap door spiders (3 Actinopodidae, 6 Migidae, 14 Nemesiidae, 21 Pycnothelidae), 8 funnel web

spiders (Hexathelidae), 4 goblin spiders (Oonopidae), 4 sand recluse spiders (Sicariidae), 13 tarantulas (Theraphosidae), 10 ant spiders (Zodariidae); closest relatives to the spiders are sun spiders (Solifugae), with 10 burrowing species within the families Mummuciidae, Daesiidae and Ammotrechidae. Related to these basal taxa are scorpions, with 50 burrowing species (most belonging to the genera *Bothriurus* sp. and *Brachistosternus* spp.). Higher taxonomic groups with burrowing species are beetles, with 157 species including 63 darkling beetles

(Tenebrionidae), 13 ground beetles, (Carabidae), 11 trogids (Trogidae), 29 dung beetles and 41 scarab beetles (both of Scarabaeidae); 13 bees (Colletidae); 5 wasps (Crabronidae and Sphecidae); 23 ants (Formicidae); 1 ant lion (Myrmeleontidae); 1 termite (Rhinotermitidae) and 1 cicada (Cicadidae) species (Übernickel et al., 2020). Bétard (2020) mentions additional orders and families of insects that burrow on a global scale. Of these (Blattodea, Dermaptera, Diptera, Embioptera, Ephemeroptera, Lepidoptera, Mecoptera, Megaloptera, Odonata, Orthoptera,

Plecoptera, Trichoptera), however we did not find supporting literature for equivalent Chilean species. The following sections provide details concerning the findings summarized separately for vertebrates and invertebrates in Chile, subdivided into the geographic distribution in Chile, composition and spatial occurrence at study sites and their burrow characteristics.

### 3.1.1 Burrowing vertebrates

The distribution of Chilean burrowing mammal species varies with latitude and altitude along Chile (Fig. 1 D), with most species having restricted distributions in distinct parts of the country. A maximum of 18 sympatric species were identified around 35° S. Most diversity is present at the mediterranean zone (central Chile) with commonly 11 to 16 species per grid cell. Trends in the species distribution are gradual with latitude and fewer species are present at higher elevations, i.e. the Andes (Fig. 1 A). Similarly, species number decreases with

decreasing mean annual temperature, i.e. Patagonia in the south (MAT, Fig. 1 B). Finally, species numbers decrease with decreasing mean annual precipitation, i.e. northern arid lowlands (MAP, Fig. 1 C).

Other burrowing vertebrates in addition to mammals includes birds and reptiles. Among the burrowing birds only the burrowing owl (*Athene cunicularia*) and the parrot "Tricahue" (*Cyanoliseus patagonus*) are identified. These species have distributions at the Chilean main land; Chilean seabirds, i.e. burrowing penguins and petrels, were

excluded from the list, as in general they are exclusively present on islands (Chester, 2008; Figueroa and Stucchi, 2008; Reyes Arriagada et al., 2013; Zavalaga and Alfaro-Shigueto, 2015). Of the three burrowing reptile species, two are *Liolaemus* species. Chile has 99 known species of *Liolaemus* sp. (Ruiz de Gamboa, 2020), but only for the two *Liolaemus* sp. the burrowing behavior was found documented. The third burrowing reptile is the Chilean Racerunner *Callopistes maculatus* (Übernickel et al., 2020) with a distribution from just north of Antofagasta to

Cauquenes (Contreras et al., 2020).

The species composition in the four study sites is composed of 27 out of the total 45 Chilean burrowing vertebrates (Table S1). The NP Pan de Azúcar study area has 13 documented (and up to 15, including burrowing species potentially present) burrowing vertebrate species, 13 (up to 17) are present in Santa Gracia, 14 (up to 17) in NP La Campana and 11 (up to 15) in NP Nahuelbuta. Very few species are present along the entire studied gradient,



i.e. two foxes, *Lycalopex griseus*, *L. culpaeus*, two grass mice, *Abrothrix olivaceus*, *A. longipilis*, Darwin's leaf eared mouse*, Phyllotis darwini*, and the introduced common house mouse, *Mus musculus*.

The knowledge of the geographic distribution and density of burrowing vertebrates in Chile is sparse for most species, with no further details other than the morphometric description of the animals available (Übernickel et al., 2020). Most details are available for some rodents, especially for the cururo, *Spalacopus cyanus*, and degu,

*Octodon degus.* Within the known burrowing species, the density of individuals varies considerably on a local level, with most species occurring in densities usually less than one individual ha$^{-1}$, up to an approximate 300 individuals ha$^{-1}$ under favorable, time restricted circumstances associated with (for example) El Niño Southern Oscillation (ENSO) events, see discussion. Mammal group sizes vary from a solitary lifestyle, usual for armadillos, skunks and foxes, to groups of more than 15 to 20 individuals for most rodents. These rodent colonies generally

inhabit multi-entrance burrows (Iriarte W., 2007; Kolb, 1985; Pearson, 1988; Redford and Eisenberg, 1992; Schmid-Holmes et al., 2001; Shepherd and Ditgen, 2013). The burrow entrances are usually clumped in a small area (Iriarte W., 2007; Pearson, 1984, 1988, 1951), with larger spacing between complex burrows. There are usually less than 2 colonies ha$^{-1}$ (Cofré and Marquet, 1999; Patton et al., 2015; Redford and Eisenberg, 1992).

Mammal burrow dimensions are highly variable and for most species the areal extent of burrows is not reported.

The following provides an overview of the known basic dimensions available for Chilean species (Übernickel et al., 2020). The largest reported burrows are of armadillos and skunks, with tunnel lengths of up to 4 m and tunnel diameters of up to 30 cm. The longest reported tunnel lengths are of up to 49 m and maximum burrow depths of 1.2 m for *Ctenomys maulinus brunneus*. The areal extent of burrow systems varies between 20-30 m$^2$ for *Chelemys macronyx*, to over 200 m$^2$ for *Ctenomys magellanicus*, or 300 m burrow lengths for *Ctenomys opimus*. Known

burrowing depths for Chilean mammals reach 3 m for skunks (*Conepatus*), 1.5 m for armadillos (Chlamyphoridae), at least 80 cm for rabbits (*Oryctolagus*), 75 cm for tuco-tucos (*Ctenomys*), 60 cm for mole rats (*Chelemys*), degus (*Octodon*) and cururos (*Spalacopus*), 28 cm for cavies and 25 cm for rock rats (*Aconaemys*).

The large group of burrowing small mammals, including rodents and insectivores, vary in their tunnel diameter between 2 and 11 cm. By diameter the entrances are roughly classifiable to distinct groups. The largest rodent-

made entrances with diameters of 7 to 10 cm are made by tuco-tucos, *Ctenomys* sp., rats *Rattus* sp., rock rats, *Aconaemys* sp. and degu, *Octodon* sp. Intermediate entrances with diameters of 4 to 7 cm are made by mole rats, *Chelemyx* sp., the rabbit rat, *Reithrodon* sp., the cururo, *Spalacopus cyanus,* and mice of several genera like *Eligmodontia, Euneomys, Geoxus, Phyllotis*, with specific tunnel diameters of 6 cm only reported for *G. valdivianus*. The smallest entrance diameters of 2.5 to 4 cm are made by grass mice, *Abrothrix* sp. / *Akodon* sp.,

the mouse, *Loxodontomys* sp. and the house mouse, *Mus musculus*.

For non-mammal burrowing vertebrates information on burrow characteristics is scarce. Concerning lizards, field observations reveal that burrows of the lizard *Callopistes* vary considerably in size, and frequently they occupy burrows made by *Spalacopus cyanus* or other rodents. The observed entrance diameters is 10 to 15 cm, 0.6 to 2 m for tunnel lengths and they have also been observed to shelter under rocks (*pers. com.* A. Cortés Maldonado, J. P.

Castillo). Regarding birds, burrowing parrots make entrances that vary largely in size, 14-49 cm horizontally and 8-25 cm vertically.

### 3.1.2 Burrowing invertebrates

The distribution of burrowing invertebrate species is heterogeneous along the north to south extent of Chile (Fig. 2). To summarize, the burrowing animal diversity is highest in the semi-arid "Norte Chico" (small north) of Chile,



followed by the mediterranean area. The humid-temperate area adjacent to the mediterranean area towards the south also contains considerable diversity. Further to the north (central Atacama Desert) and further to the south (Patagonia) comparatively small numbers of burrowing species are registered. In detail, of the total 345 species identified, the largest number (131) is present in the region de Coquimbo (IV). Other regions with high diversity in burrowing invertebrates are the regions de Valparaíso (87), del Biobío (81), Metropolitana (75) and de Atacama

(71, Fig. 2). An intermediate diversity is present at the region de Maule (69), region de Araucanía (66), region de Ñuble (63), region de Los Lagos (57), region de L.G.B. O'Higgins (56) and region de Los Ríos (41). In the following, we summarize for the species with the respective information available (Übernickel et al. 2020).

The distribution of the basal taxa, spiders and scorpions, cover nearly all Chile (Fig. 2). Generally, burrowing spiders (Arachnida) are distributed from the region de Antofagasta (II) southwards, most are distributed in central

to southern Chile. Trap door spiders and tarantulas are more common in mediterranean and humid-temperate sites, being more of an exception in arid and semi-arid sites. Ant and camel (sun) spiders have distribution over most of Chile. Of the Nemesiidae (family of trap door spiders), the genus *Lycinus* spp., consisting of 8 species, has a distributional range in the northern half of Chile, from the regions Antofagasta to Valparaíso (II-V). Solifuges, are present only in the semi-arid and adjacent regions. Scorpions as a group have wide elevational distributions, from

the coastal desert to the high Cordillera of the Andes (0 up to 4,500 m a.s.l.). For individual scorpion species the pattern differs: single species exist either at (semi-) arid ranges, or at mediterranean to humid-temperate ranges. The scorpion with the broadest distribution in Chile, from the region of Antofagasta (II) to the region of Valparaíso (V), is *Caraboctonus keyserlingi*.

The distribution of the taxonomically higher and (mostly) volant groups of beetles and hymenopterans cover all

of Chile, but subgroups are restricted to specific climatic areas. Here we summarize for beetles, bees and ants, for other subgroups information is not available. Darkling beetles (*Gyriosomus* sp.) inhabit arid and semi-arid sites only, whereas ground beetles, scarab beetles and trogids have wider distributions. Dung beetles are present in humid areas. *Praocis* (*Mesopraocis*) beetles inhabit mainly the Chilean Coastal Desert. Burrowing bees are generally distributed from the north to the central region, such as the genus *Caupolicana* (Colletidae), is distributed

from Arica to the Araucanía. Ants, represented in Chile by ca. 65 species (Cuezzo, 2007; Johnson and Moreau, 2016; Snelling and Hunt, 1975) are distributed over all Chilean biomes from sea level to 2500 m a.s.l. (Ipinza-Regla et al., 1983). A maximum of 47 sympatric species is present in Central Chile (antmaps.org, also for genus distributions). Of burrowing species, the large sized "hormigones" (*Camponotus*), is one of the most abundant and widely distributed genera in Chile (Snelling and Hunt, 1975). The fire ant genus *Solenopsis* includes *S. gayi,* the

most common native ant in Chile with the widest distribution. Of the Neuropterans, antlions are present in northern (Meserve et al., 2016; Miller and Stange, 2016) and central Chile.

Species compositions of burrowing invertebrates are unknown for most parts of Chile. We approximated the compositions by counting all burrowing species with published presence within the regions of the study sites. "Likely present" species according to current knowledge were also included. We figure the following species

diversity (Table S2): In NP Pan de Azúcar there are 58 burrowing invertebrate species (10 spiders, 2 solifuges, 8 scorpions, 26 beetles, 3 bees, 3 wasps, 5 ants, 1 cicada), the lowest diversity across the four research sites. Compared to the other sites this arid site hosts notably the fewest spider, beetle and ant species. In Santa Gracia the highest number of burrowing invertebrates is present, with 93 species (27 spiders, 3 solifuges, 3 scorpions, 40 beetles, 7 bees, 3 wasps, 9 ants, 1 cicada). Compared to the northern arid site, at this semi-arid site the numbers of

spider species, beetles, bees and ants are nearly doubled. Towards the south, the overall number of burrowing



species remains similar, but with shifts within the taxonomic groups. In NP La Campana 101 burrowing species are likely present including 32 spiders, 5 solifuges, 6 scorpions, 28 beetles, 7 bees, 5 wasps, 15 ants, 1 termite, 1 antlion, and 1 cicada. Compared to the previous semi-arid site, here again an increase in the number of spider species number is observable. The scorpion diversity is doubled, beetle diversity is halved and there is noticeably

more ant diversity. At the southernmost and humid-temperate site NP Nahuelbuta 99 burrowing species are estimated present including 26 spiders, 10 scorpions, 41 beetles, 3 bees, 1 wasp, 17 ants and 1 cicada. Compared to the previous mediterranean site NP La Campana, the total number is very similar, but there are shifts towards more scorpion and beetle diversity. Burrowing termites and antlions are not known to occur this far south.

Known details of the spatial distribution of burrowing Chilean invertebrates are very scarce. Habitat preferences,

e.g. specific substrates species are associated to, are registered for very few species (Übernickel et al., 2020). Details of densities are only recorded for tarantulas (Theraphosidae), outside of Chile they are estimated to have a density of 0.07 to 0.65 per m$^2$ (Pérez-Miles et al., 2005). Some Chilean genera of tarantulas (e.g. *Euathlus*, *Grammostola* and *Paraphysa*) are likely to have very reduced densities, compared to undisturbed conditions, as their trade for the pet industry was banned only in 2018 (Servicio agrícola y Ganadero, SAG, 2018).

Here, the dimensions of invertebrate burrows in Chile and presented and grouped for the basal taxa first, followed by the volant species' details. Trap door spiders make simple tunnels with diameters of 0.5 to 2 cm. Of the trap door spider family Actinopodidae, *Plesiolena bonneti* burrows in 45° inclination to horizontal. A sister species, *P. jorgelina*, produces burrow tunnels with diameters of 2 cm and to depths of 25 cm. The trap door spider family Nemesiidae burrow with diameters between 0.7 and 1.6 cm and to a maximum depth of 30 cm. Of this family

*Lycinus* sp. excavate tunnels of approximately 1.0-1.5 cm in diameter with more than 10 cm depth and the burrow walls are coated with silk. Of the fourth family of trap door spiders, Pycnothelidae (ex-Nemesiidae), *Acanthognathus* sp. make round burrows with estimated entrance diameters of 2 to 2.5 cm, also coated with very fine silk. Tarantulas usually burrow vertically approximately 1 m in depth and coat the inside of their burrows with silk (Perafán and Pérez-Miles, 2014). Among scorpions, species of the family Bothriuridae construct simple

burrows of 25 to 50 cm depth, reaching up to 1 m depth occasionally. Female solifuges at locations outside of Chile are described to burrow down to 10 to 20 cm for oviposition.

Burrow dimensions of higher taxa and (mostly) volant groups of beetles, bees, wasps, ants and cicadas are summarized in the following. For other subgroups information is not available. Chilean dung beetles reach burrow depths of 40 to 60 cm. The females of *Gyriosomus* (Tenebrionidae) and trogids reach burrow depths of 20 cm.

*Praocis* (*Mesopraocis*) sp. are usually found within the uppermost 30 cm of the substrate, but have also been found close to 2 m depth (*pers. obs.* J. Pizarro-Araya). They lay eggs on surfaces or at depths between 5 and 10 cm. Both burrowing bee genera, *Caupolicana* and *Cadeguala*, burrow their nests in the soil to around 50 cm depth. The burrows of *Cadeguala occidentalis* reach extensions of 120 * 170 cm with over 150 entrances. The "digger wasp" *Sphex* sp. burrows reach 33 cm depth in regions outside of Chile (Brockmann, 1980). The burrows of the cicada

*Tettigades chilensis* reach depths of 45 cm.

### 3.1.3 Published excavation rates of burrowing animals

Globally, excavation rates of single vertebrate and invertebrate species are published for specific sites (Table 1). Among vertebrates, the globally published excavation rates vary over several orders of magnitude, with the largest excavation rate reported as 146 m$^3$ ha$^{-1}$ yr$^{-1}$ for the mountain pocket gopher in a study in the USA. Intermediate

excavation rates of 20 to 60 m$^3$ ha$^{-1}$ yr$^{-1}$ are more commonly reached by vertebrates, with the highest values by





Botta's pocket gopher (56.2 m$^3$ ha$^{-1}$ yr$^{-1}$), the house mouse (20.6 m$^3$ ha$^{-1}$ yr$^{-1}$) and moles and voles together (20 m$^3$ ha$^{-1}$ yr$^{-1}$). A range of excavation rates from 10 to 19 m$^3$ ha$^{-1}$ yr$^{-1}$ is reached by the European mole in Russia and the arctic ground squirrel. Excavation rates up to 10 m$^3$ yr$^{-1}$ are reported for badgers, bettongs (rat kangaroos), cururos (rodents) and porcupines. In contrast, small excavation rates are reported for rabbits (0.68 m$^3$ ha$^{-1}$ yr$^{-1}$). Similarly,
the globally published excavation rates for invertebrates also vary significantly (Table 1) with the highest excavation rates reached by ants with 53.33 m$^3$ ha$^{-1}$ yr$^{-1}$ and earthworms at ~33.83 m$^3$ ha$^{-1}$ yr$^{-1}$. Ants and earthworms reach the same order of magnitude and even higher values than vertebrates' intermediate excavation rates, see above. In comparison, low maximum volumes are published for termites, which range between 0.25 and 2.5 m$^3$ ha$^{-1}$ yr$^{-1}$.




Table 1: Global animal excavation rates from literature. When the original data was mass data, an intermediate bulk density of 1.2 g cm⁻³ following Amelung et al. (2018) was assumed for conversion

| Common name | Latin name | Study site | Excavation rate [m³ ha⁻¹ yr⁻¹] | | | Publication |
|---|---|---|---|---|---|---|
| | | | mean | min. | max. | |
| Badger, American | Taxidea taxus | USA | 4.25 | | | Eldridge 2004 |
| Badger, European | Meles meles | UK | | 0.012 | 0.03 | Coombes & Viles 2015 |
| Bettong, brush-tailed | Bettongia penicillata | Australia | 3.33 | | | Garkaklis et al. 2004 |
| Bettong, brush-tailed | Bettongia penicillata | Australia | 1.33 | | | Garkaklis et al. 2004 |
| Coruro | Spalacopus cyanus | Chile | 7.37 | | | Contreras et al. 1993 |
| Gopher | Citellus pygmaeus | Northern Caspian Lowland (Russia / Kazakhstan) | 1.80 | | | Abaturov 1972 |
| Gopher, Botta's pocket | Thomomys bottae | USA | | 3.23 | | Black & Montgomery 1991 |
| Gopher, Botta's pocket | Thomomys bottae | USA | | | 3.01 | Black & Montgomery 1991 |
| Gopher, Botta's pocket | Thomomys bottae | USA | | 22.31 | 56.20 | Gabet 2000 |
| Gopher, Botta's pocket | Thomomys bottae | USA | 23.78 | | | Cox 1990 |
| Gopher, Mountain pocket | Thomomys monticola | USA | 146.20 | | | Ingles 1952 |
| Gopher, northern pocket | Thomomys talpoides | USA | 9.45 | | | Ellison 1946 |
| Gopher, pocket | Geomys breviceps | USA | | 0.74 | 14.58 | Buechner 1942 |
| house mouse | Mus musculus | Sub-Antarctic | 20.60 | | | Eriksson & Eldridge 2014 |
| Mole | Talpa europaea | Poland | | 0.83 | 2.50 | Jońca 1972 |
| Mole | Talpa europaea | Russia | | 3.25 | 15.50 | Abaturov 1972 |
| Mole & Vole | (Talpa sp. & Microtus sp.) | Luxembourg | 20.00 | | | Imeson 1976 |
| Mole, europaean | Talpa europaea | Netherlands | 3.66 | | | Wijnhoven et al. 2006 |
| Porcupine, Cape | Hystrix africaeaustralis | South Africa | 2.2 | | | Bragg et al. 2005 |
| Porcupine, Indian crested | Hystrix indica | Israel | | 0.12 | 2.06 | Shachak et al. 1991 |
| Rabbit | Oryctolagus cuniculus | Netherlands | 0.68 | | | Rutin 1992 |
| Squirrel, Arctic ground | Citellus undulatus | Canada | 15.84 | | | Price 1971 |
| Squirrel, columbian ground | Spermophilus columbianus c | Canada | | 0.93 | 1.13 | Smith & Gardner 1985 |
| Vole, common | Microtus arvalis | Netherlands | 0.02 | | | Wijnhoven et al. 2006 |
| Ants | Atta vollenweideri | Argentina | 9.17 | | | Bucher & Zuccardi 1967 in Lobry de Bruyn & Conacher 1990 |
| Ants | Formica exsectoides | USA | 14.40 | | | Salem & Hole 1968 |
| Ants | Lasius flavus | UK | 6.87 | | | Waloff & Blackith 1962 in Bétard 2020 |
| Ants | Pogonomyrmex badius | USA | 53.33 | | | Tschinkel, 2015 |
| Ants | Solenopsis invicta | USA | 1.33 | | | Betard, 2020 and Paton et al 1995 |
| Bee, alkali | Nomia melanderi | (USA) | 0.01 | | | Cane 2003 in Butler et al. 2003 |
| Beetles | Copris tullius, Pinotus carolinus | USA | 0.13 | | | Lindquist 1933 in Bétard 2020 |
| Beetles | Peltrotupes young | USA | | 0.02 | 3.08 | Kalisz and Stone 1984 |
| Earthworms | Allolophora nocturna | Luxembourg | 15 | | | Hazelhoff et al. 1981 |
| Earthworms | Lumbricus sp. | UK | | 14.08 | 33.83 | Darwin 1881 in Bétard 2020 |
| Termites | Cubitermes sp. | Africa | 2.5 | | | Aloni & Soyer 1987 in deBruyn & Conacher 1990 |
| Termites | Macrotermes bellicosus | West Africa | 1.04 | | | Nye 1955 in Bétard 2020 |
| Termites | Macrotermes sp. | West Africa | | 0.25 | 0.88 | Goudie 1988 in Bétard 2020 |
| Termites | Trinervitermes trinewoides | West Africa | 0.30 | | | Nel and Malan 1974 in Bétard 2020 |



### 3.1.4 Maximum burrow depths of vertebrates and invertebrates

Globally, the documented maximum depth of animal burrows is 6 m (gopher tortoises and aardvark dens, Platt et al., 2016). Among invertebrates, the basal taxa of scorpions and spiders reach burrow depths of 70 and 60 cm, respectively (Framenau and Hudson, 2017; Talal et al., 2015). Ants' nests are documented up to depths of 4 m, and cicadas reach depths of 2.5 m (Bétard, 2020; Tschinkel, 2004).

Chilean burrowing mammals are described to reach maximum burrow depths of 3 m for skunks, 1.5 m for
armadillos, > 80 cm for rabbits, < 75 cm for tuco-tucos, ~ 60 cm for mole rats (*Chelemys*), degus and cururos, ~ 28 cm for cavies and < 25 cm for rock rats (*Aconaemys*). For invertebrates maximum burrowing depths are estimated to range from 100 cm for scorpions to 30 cm for spiders. Higher taxa burrow depths are 60 cm for beetles, 52 cm for bees and 45 cm for cicadas.

### 3.2 Study: Observed animal burrows in Chilean study areas

A total of 1,590 burrow entrances were found within the plots and measured throughout the three field seasons in 2016 to 2018 (yellow stars, Fig. 1, Fig. 3). During all field visits animal burrow entrances were identified at all sites and in all plots. In general, there are two types of animal burrows including: single and small entrances, like those of most invertebrates, or clumped multiple and larger entrances, typical of most rodent burrows. To simplify analysis, burrow entrances were separated into two groups using a diameter of 2.5 cm as threshold. The threshold
is based on the relation of a burrow entrance diameter to the hosts' body width (e.g. Gabet et al., 2003; Vleck, 1981). The smallest entrance diameter reported for mammals is 2-3 cm for *Mus musculus* (house mouse, Table S1). Here we follow the approach of Kelt et al. (2004) and define a threshold diameter of 2.5 cm to include grass mice. Given this, entrance diameters < 2.5 cm were classified as from "small animals", mostly invertebrates, and ≥ 2.5 cm for "large animals", mostly mammals (Fig. 9).

Field observations in Santa Gracia revealed the presence of the lizard *Callopistes maculatus* identified by characteristic feces adjacent to burrow entrances, located next to a dry riverbed. As for burrowing birds, the owl *Athene cunicularia*, and the parrot *Cyanoliseus patagonus*, were observed. For invertebrates, burrow entrances of approx. 1.5 cm in diameter and the characteristic exuviae of the nymphs of the cicada *Tettigades chilensis* were found (*pers. obs.* K. Übernickel & J. Pizarro-Araya in Santa Gracia). In Quebrada de Talca, near Santa Gracia, a
large ground nest of Hymenopterans with a surface extension of approx. 1 m$^2$ was observed. In NP La Campana, burrowing behavior of a female Mummuciidae was observed.

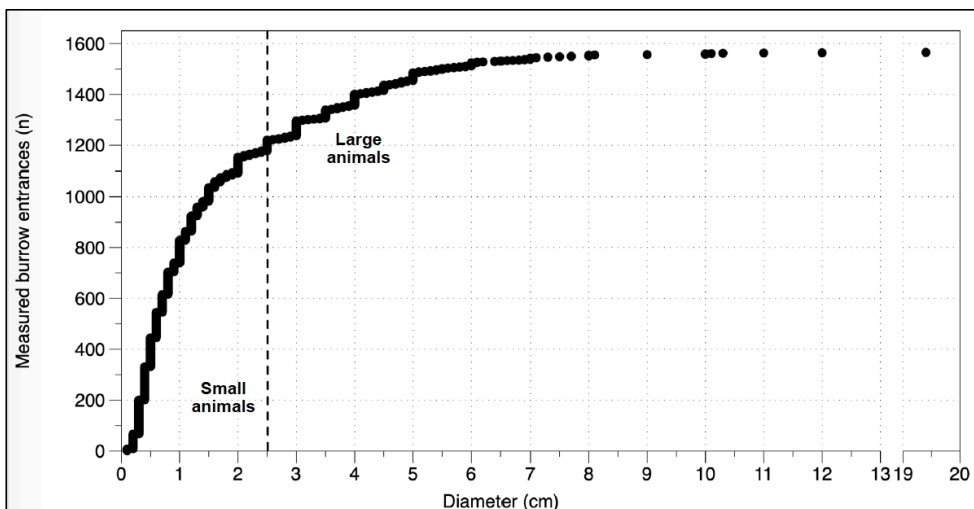

Figure 9: Number of measured entrances sorted by diameter. The dashed vertical line indicates the 2.5 cm diameter value of classification between small animals and large animals, $n_{total}$=1565, $n_{small\ animal}$=1177; $n_{large\ animal}$=388


### 3.2.1 Burrow entrance distribution

The number of entrances found varies per plot between 1 and 108 (median: 24, Fig. 5). The majority of burrow entrances, approximately three-quarter of all entrances, are originated by small animals. Large animal entrances were found on all slopes. Our observations do not indicate a pattern across top and bottom positions of the plots.

Across slopes, more entrances are found on the north-facing slopes than on the south-facing slopes in the two northern sites. In the mediterranean climate of the NP La Campana, the number of registered entrances is the same across both slopes. For the southernmost NP Nahuelbuta available data is too sparse to allow any conclusion from visual inspection. Along the climate gradient, more entrances are found in arid (n=443) or semi-arid sites (n=624), compared to the mediterranean (n=362) or humid-temperate study areas (n=136).

### 3.2.2 Relationship between burrows and climate

The number of entrances per slope and along the climate gradient could be related to incoming solar radiation. The number of entrances is plotted together with the calculated summer and winter radiation energy on north and south-facing slopes (Fig. 8). The maximum of the solar radiation at noon throughout a year varies with latitude, and the amount decreases from 26° S (PA) to 38° S (NA). During the summer, at latitudes >30° S (Santa Gracia)

the radiation is higher on north-facing slopes than on south-facing slopes. During winter, there is higher solar radiation on all north-facing slopes, compared to south-facing slopes. The amount of burrow entrances decreases from the semi-arid site (Santa Gracia) towards the south. Within each site, the majority of the small animal entrances is located in accordance with the higher radiation on north-facing slopes. The number of large animal burrow entrances is also higher on north-facing slopes at the arid and semi-arid sites. In contrast, at the

mediterranean and humid-temperate sites, large animal entrance numbers are higher at the south-facing slopes. Similar to the incoming solar energy, MAT is high at the arid site, peaks at the semi-arid Santa Gracia site, and then decreases towards the south (Fig. 10). The MAT in Chile varies from desert maximum values of 20° C in the north to temperatures around 0° C or below in the south of Chile. Small animal and large animal entrances follow





the same pattern decreasing towards the south with decreasing MAT. MAP counteracts this pattern (Fig. 10). The
MAP in Chile varies from near zero in the north to above 200 mm in the humid south.

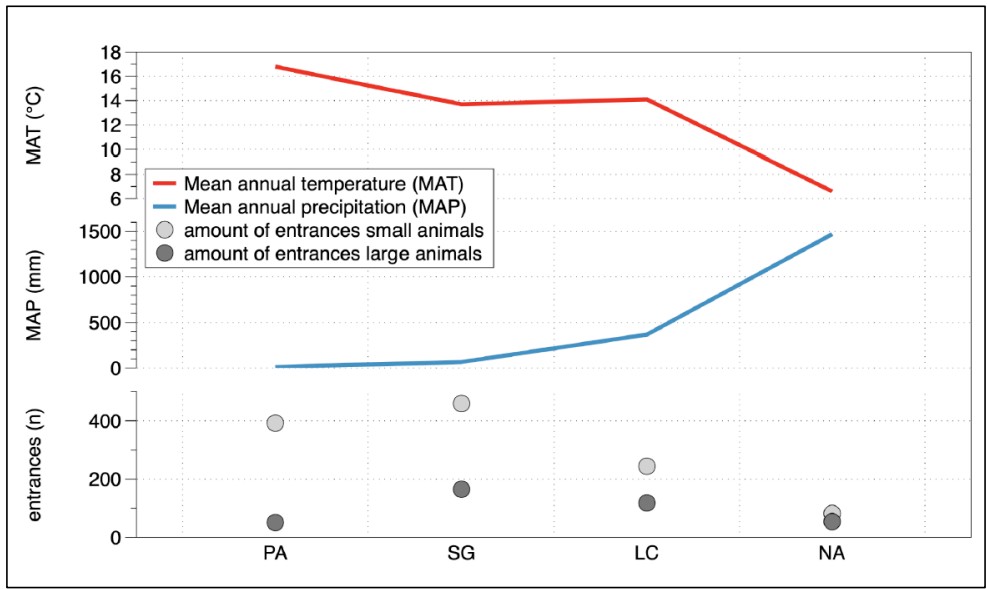

Figure 10: Mean annual temperature (MAT) and mean annual precipitation (MAP) and animal burrow entrance distribution
along the climatic gradient; the depicted data is pooled from all three field campaigns 2016, 2017 and 2018; MAT & MAP
from Fick and Hijmans (2017); Abbreviations: PA= Pan de Azúcar, SG= Santa Gracia, LC = La Campana, NA = Nahuelbuta,
*note that in PA the sum of the south-facing plot entrances is of one plot, and the north-facing entrances sum of three plots.*

### 3.2.3 Burrow dimensions

The burrow dimensions measured were entrance diameter, minimum tunnel length and tunnel orientation. The
correlation of the diameter and the minimum tunnel lengths are positive for both groups (Fig. 4), steeper for small
animals (slope: 5.84) and shallower for large animal entrances (slope: 3.83). The longest straight tunnel length
was 47 cm with a diameter of 3.5 cm. For invertebrates the longest detected tunnel length was 16 cm with 2 cm
diameter. The minimum burrowed depth, calculated from minimum tunnel length and tunnel orientation, has a
mean value of < 6 cm depth (range of mean values for small and large animals per plot: 1.6 – 11.6 cm, Fig. 6).
Maximum values of calculated burrowing depth are < 26 cm for all sites and slopes. For reference, we plotted soil
thickness (Oeser et al. 2018) and mobile layer thicknesses from Schaller et al. (2018) from the same sites (Fig. 6).
Both the soil and mobile layer thicknesses have values around 20 cm depth in the north and increase towards the
south reaching values around 60 to 90 cm depth.

### 3.2.4 Minimum excavation rates

Minimum excavation rates were calculated from tunnel volumes (Fig. 7). The species independent minimum
excavation rates increase from north to south (calculated including all animals summed for both slopes: PA: 0.34
$m^3$ $ha^{-1}$ $yr^{-1}$; SG: 0.56 $m^3$ $ha^{-1}$ $yr^{-1}$; LC: 0.93 $m^3$ $ha^{-1}$ $yr^{-1}$; NA: 0.09 $m^3$ $ha^{-1}$ $yr^{-1}$), except for the southernmost site.
Even through the number of entrances generated by small animals is always larger, the excavation volumes
generated by large animals is about 1.5 magnitudes larger than small animal entrances on the same slope.





Comparing across slopes in each study area the sum of minimum excavation rates (Fig. 7) are larger on the north-facing slope in NP Pan de Azúcar (south-facing: 0.007 m³ ha⁻¹ yr⁻¹; north-facing 0.675 m³ ha⁻¹ yr⁻¹) and also in Santa Gracia (south-facing: 0.135 m³ ha⁻¹ yr⁻¹; north-facing: 0.988 m³ ha⁻¹ yr⁻¹). In La Campana, the contrary pattern is visible, i.e. a larger excavation rate on the south-facing slope (south-facing: 1.459 m³ ha⁻¹ yr⁻¹; north-facing 0.397 m³ ha⁻¹ yr⁻¹). At the southernmost site, NP Nahuelbuta, the excavation rate is similar on both slopes (south-facing: 0.090 m³ ha⁻¹ yr⁻¹; north-facing: 0.088 m³ ha⁻¹ yr⁻¹). Focusing on the subgroup of small animal minimum excavation rates, they are larger on the north-facing slopes at the three northern sites. For large animal minimum excavation rates a clear pattern is not observed with the available data.

## 4 Discussion

### 4.1 Literature compilation on Chilean burrowing animals

In this literature review we focus on an inventory of burrowing vertebrates and invertebrates along the climate and ecological gradient in Chile. Currently known burrowing species vary considerably along Chile, mammal diversity was highest for the mediterranean biome, invertebrates peaked in their diversity in the semi-arid areas.

#### 4.1.1 Relationship between burrows and climate

Previous work suggested the composition of burrowing animals is related to a specific site's climate (Crawford et
al., 1993; Paton et al., 1995). This finding is confirmed in our literature review. For mammals, we found a maximum of 18 sympatric species in central Chile with a mediterranean climate in the lowland. Some burrowing species are adaptable to varying climate conditions such as the Chilean rodent cururo *Spalacopus cyanus* (Begall and Gallardo, 2000; Contreras, 1986), that adapted its body size and behavior to the local conditions, to adapt over a larger climatic gradient. According to their known distribution, foxes, *Lycalopex griseus*, *L. culpaeus* and small
mammals *Abrothrix olivaceus*, *A. longipilis*, *Phyllotis darwini*, and *Mus musculus* seem also very flexible in the climate in which they live. For invertebrates, the quantity of sympatric species has a maximum of 131 species in the region of Coquimbo (IV), part of the "Norte Chico" semiarid climate (Fig. 2). Some invertebrate groups are restricted to specific biomes such as darkling beetles in the arid and semi-arid regions, solifuges in semi-arid regions, trap door spiders or tarantulas in mediterranean areas, or dung beetles in humid settings. Fewer species,
mostly ants, ant and camel (sun) spiders, as well as scarab beetles have a very wide distribution across several biomes.

#### 4.1.3 Burrowing animal excavation rates

The published global species specific excavation rates reach maximum reported values around 146 m³ ha⁻¹ yr⁻¹ (Mountain pocket gopher, Ingles, 1952) (Table 1). More common are rates up to 25 m³ ha⁻¹ yr⁻¹ for vertebrates.
Interestingly, values for invertebrates reach the same order of magnitude, especially earthworms at 15-34 m³ ha⁻¹ yr⁻¹ (Darwin, 1881; Hazelhoff et al., 1981), or ants with excavation rates of 53.33 m³ ha⁻¹ yr⁻¹ (Tschinkel, 2015). For most burrowing invertebrates data about excavated volumes remains unknown. The species-specific values reported in literature are potentially higher than the average burrowing activity, as most study sites were selected because they had a high number of burrows of the target species present (rather than doing a plot scale inventory
as in this study). Study outcomes may also vary to a certain degree by different sampling methods, as e.g. burrow volumes may reach more than 20 times the volume of the mounding at the surface (Bétard, 2020). For a more





general discussion readers are referred to previous work by Hausmann (2017) and Wilkinson (2009), and Smallwood and Morrison (1999a) on gopher burrows.

### 4.2 Observations of animal burrow entrances in four Chilean study areas

This study consolidated previous results from literature regarding burrowing animals' composition and spatial occurrence variation across the four studied biomes along a climate gradient. In addition, our field study areas (Fig. 3) documented the zoogeomorphic effects on hillslope mass transport at the animal community level and along the climate gradient. We found the distribution of burrow entrances is heterogeneous for both vertebrates and invertebrates burrowing species within a biome as well as along the climate gradient. More burrow entrances

were found at north-facing slopes than on opposing slopes in two of the four biomes, and the highest excavation rates were found in the semi-arid and the mediterranean biomes. Links of burrow activity with soil thickness, MAT, MAP and solar radiation are discussed in the following.

### 4.2.1 Distribution of burrow entrances

Varying soil thickness at top and bottom slope positions of hillslopes could influence the burrow entrance

distribution. A thicker soil layer at bottom slope positions potentially provides more substrate and nutrients for vegetation to grow bulbs and rhizomes in, that in turn may provide more favorable food conditions for burrowing herbivores. The soil thickness measurements on south-facing slopes of our study areas increased from arid to humid-temperate sites from approximately 20 cm to 90 cm and also increased from top to bottom measurements within the slopes (Oeser et al. 2018, Fig. 6). Despite these findings, we did not find a difference in the distribution

of burrow entrances within top and bottom slope positions (Fig. 5). Soil thickness appears not to be a limiting factor for burrow entrance distribution between top and bottom slope.

Burrow entrance distribution could also be influenced by the incoming solar energy, that is higher on north-facing slopes in comparison to south-facing slopes in the southern hemisphere and has an effect on the available energy and also on water balances (Gallardo-Cruz et al., 2009). The effect of the hillslope aspect (and hence available

solar energy) on animal excavation activity has been addressed in previous work. In the northern hemisphere in an alpine region in Canada the total sediment displacement of animals has been highest on east and south-facing slopes, compared to north- and west-facing ones (Hall and Lamont, 2003). Furthermore, in a Chilean animal trap study at a semi-arid north-central site, north-facing slopes have had the doubled abundance of small mammals, mostly burrowing species, compared to south-facing slopes (Jiménez et al., 1992). Our findings of a higher number

of entrances on north-facing slopes than on south-facing slopes were consistent with both studies in the arid and semi-arid biomes (Fig. 5).

### 4.2.3 Differences in the distribution of large and small animal burrows

We observed small animal burrows to be mostly distributed evenly over the plot surface areas, large animal burrows were mostly distributed in a clumped manner. Furthermore, large animal burrow entrances were present

on all slopes but not in all plots. We adopted to this distributional difference by separation of the data into two groups by their burrow entrance diameter (see results section 3.2) to gain a better understanding of the patterns. If large animal burrow entrances were not present in the sampled plot, it provides no indication about their absence or presence and the resulting effect on a hillslope, as an area *between* colonially used burrows may have been selected. When large animal burrow entrances were present in a plot, it resulted in comparatively high impact



which may overprint all volumes excavated by small animals that were up to three-quarters of locally counted entrances (Fig. 7).

In addition, even single large animal dens, i.e. fox dens, may overprint all other burrowing activity. In NP Pan de Azúcar, adjacent to the plots within a 90 m radius, we observed larger holes that gave the impression to be dens or burrow entrances. 21 of these structures were measured in 2017, which were oriented in all directions. At the

bottom of the slopes they were excavated farther than further uphill. Occasionally feces were found in front of the entrances. The diameter of the structures was on average 15 cm (range: 7-57 cm), height was on average 14 cm (range: 4-45 cm), and the excavated depth was on average 34 cm (range: 10-76 cm).

### 4.2.4 Excavated depths

Animal groups have different burrowing depths. By burrowing, animals are able to maintain favorable conditions

of temperature and humidity compared to above surface conditions (e.g. Bétard, 2020). Described burrowing depths in literature for Chilean burrowing animals ranges for vertebrates from 0.25 to 3 m and for invertebrates between 0.3 and 1.0 m. The observed maximum excavated depth measured at all four study sites was < 26 cm (Fig. 6). A previous study at the same study sites had revealed an increasing depth of the mixed layer from arid to more humid conditions, with a maximum of 85 cm in NP La Campana, followed by 70 cm in NP Nahuelbuta, 45

cm in Santa Gracia and 17.5 cm in NP Pan de Azúcar (Fig. 6) (Schaller et al. 2018). The calculated minimal burrow depth reached the soil mixing depth values from literature only at the arid site. Based on the maximum burrow depths from literature for the differing taxa in Chile, we presume that real burrow depths are similar to the mobile layer depths reported in the study of Schaller et al. (2018). The observed result is likely an underestimation caused by the measurements procedure, i.e. the inability to easily determine the full burrow extent deeper than the

first curvature after the entrance.

### 4.2.5 Relationship between burrows, climate, and climate variation

A correlation between the burrowing activity and regional climate was hypothesized. As indicators of gradients in climate along the extent of Chile we investigated incoming solar radiation, MAT and MAP. The burrowing activity of both, vertebrates and invertebrates, along the climatic gradient appeared to be linked to all three variables (Fig.

8, 10). The quantity of animal burrow entrances across the study sites showed the highest activity in the semi-arid Santa Gracia study area (Fig. 3), followed by arid NP Pan de Azúcar, NP La Campana and lowest activity for NP Nahuelbuta. Because of the clumped distribution patterns of large animal entrances (see discussion above), the small animal entrances resemble a more stable indicator. Previous work has found a temperature gradient to have an effect on arthropod density (Tiede et al., 2017) with fewer species at lower mean temperatures. Our results

support this finding.

As previously stated, species groups with the largest zoogeomorphic effects vary with climate, e.g., in humid areas dominant bioturbators are earthworms (Lumbricidae), whereas in arid areas the effects of mammals are highest (Paton et al., 1995; Whitford and Kay, 1999; Wilkinson et al., 2009). In our results, the minimum excavation rates reached a maximum for invertebrates in the semi-arid Santa Gracia, and for vertebrates the highest impact was

found in mediterranean La Campana (Fig. 7). Nevertheless, a more detailed understanding of animal activity, especially within the invertebrates, is necessary to assign the impact to specific groups.

In addition to the gradual change in MAP with latitude (Fig. 1), climate variation due to ENSO cycles can generate variations in the local climate conditions. ENSO consists of increased precipitation events in periodic intervals of,



on average, of every four years and lasting 2–10 years (Cane, 1983). The temporarily increased water availability during ENSO enhances vegetation in arid regions (Gutiérrez et al., 2000; Jaksic, 2001), including rainfall triggered seed production (Gutiérrez et al., 1997). The variation in changing food availability from increased vegetation affects animal densities over several orders of magnitude (Fuentes and Campusano, 1985; Holmgren et al., 2006; Jaksic, 2001; Lima et al., 1999; Meserve et al., 1995, 2011; Milstead et al., 2007; Pearson, 1975; Previtali et al., 2009, 2010). Rodent populations are described to react with "ratadas", very high abundances of rodents, to mast

seeding production, triggered by favorable conditions for plants. This applies, for example, to the granivore rat *Oligoryzomys* sp. and *Abrothrix* sp. that feed on bamboo seeds (*Chusquea* sp.) in Chile (Boric-Bargetto et al., 2012; Gallardo and Mercado, 1999; González et al., 2000) and also to other small mammal communities in Northern Chile (e.g. Jiménez et al., 1992; Milstead et al., 2007). Epigean arthropod assemblages of the Chilean coastal desert have also reacted to an ENSO event with an augmented number of individuals in most of the

recorded taxa, with emphasis on Hymenoptera (Formicidae) and Coleoptera (Tenebrionidae) (Cepeda-Pizarro et al., 2005). The years during with data acquisition were conducted in this study did not cover an ENSO cycle, so we cannot corroborate the results of previous studies with our observations.

### 4.2.6 Minimum excavation rates

The species independent excavation rates in four biomes studied in Chile (Fig. 3) were quantified. Although the

field observations presented provide only initial estimates, they nevertheless highlight trends and patterns present at the hillslope scale. Largest total sediment volumes were displaced in the south-facing mediterranean slope (1.46 $m^3$ $ha^{-1}$ $yr^{-1}$, NP La Campana). Along the climate gradient the minimum excavation rates increased from arid to mediterranean, with a decrease at the humid-temperate site such that values were 0.34 $m^3$ $ha^{-1}$ $yr^{-1}$ for the arid site, 0.56 $m^3$ $ha^{-1}$ $yr^{-1}$ for the semi-arid site, 0.93 $m^3$ $ha^{-1}$ $yr^{-1}$ for the mediterranean site, and 0.09 $m^3$ $ha^{-1}$ $yr^{-1}$ for the

humid-temperate site. Compared to published species-specific excavation rates (Table 1), our Chilean study area estimates are in the range observed for invertebrate species. As discussed above, the direct comparison of calculated excavation rates with species specific rates from literature has to be evaluated with caution. On one hand, the presented animal community values are likely to present a more representative excavation rate, than the numbers of species-focused studies. On the other hand, the presented values (0.09 to 1.45 $m^3$ $ha^{-1}$ $yr^{-1,}$ Fig. 7) are

minimum excavation rates limited by the study design. The local species independent excavation rates are likely to be higher, but currently unquantifiable with available observations.

For the invertebrates, observations resulted in a range of excavation rates between 0.003 to 0.063 $m^3$ $ha^{-1}$ $yr^{-1}$ (Fig. 7). The values of identified small animal burrow entrances were likely more reliable from the arid and semi-arid sites, because of the bias in litter cover at the southern sites. The minimum excavation rates obtained were up to

one order of magnitude larger than the reported value for alkali bees as single invertebrate species (0.005 $m^3$ $ha^{-1}$ $yr^{-1}$) (Cane, 2003). The order of magnitude for excavation rates by small animals in this study are two to three orders below the excavation rates of large animals. Regardless, the number of entrances and excavated volumes of small animals also have important impacts on other factors than downhill erosion, such as soil mixing, water runoff or infiltration, bulk density and a range of further secondary effects of bioturbation (Bétard, 2020; further

discussion about this topic in Birkby, 1983; Carlson and Whitford, 1991; Jouquet et al., 2006; Lobry de Bruyn and Conacher, 1990; Thorp, 1949; Whitford, 1996).




### 4.3 Study caveats and future research needs

In our literature review large gaps of knowledge appeared regarding excavating habits in most species, distributional ranges of species, species' densities, excavated masses or volumes, and burrowing depths. Without
better knowledge of these values the quantification of downhill erosion by burrowing animals is limited. The method presented in this study is offered as an approach to deduce minimum excavation rates for different animal burrow entrances, independent of animal taxa. In the following, improvements upon the methods used here are discussed.

One issue encountered is the increasing ground cover by vegetation that occurs towards the south and reduces the
detection of entrances in our two southernmost study areas (Fig. 3). In NP La Campana the ground was covered with litter where in the southern most NP Nahuelbuta dense understory vegetation, mainly bamboo twigs and litter, covered the ground. The amount of burrow entrances is very likely to be reduced and also the minimum excavated volume is presumably underrepresented. Removal of the ground litter would imply a massive disturbance of the plot surface, that would reduce the probability of repeated observation of undisturbed burrowing behavior.

A second issue for consideration is the seasonality of a species' activity needs to be taken into account when estimating a species' abundance and effect on burrowing. Individual species' activity patterns in Chilean vertebrates and invertebrates may result from species specific reproduction cycles (e.g. Yunger et al., 2002). Different groups have activity peaks over different times within seasons, making repeated measurements during a year time span favorable. Furthermore, repeated measurements over several year variations in climate variations
(e.g., ENSO cycles) and vegetational change are recommended, and dry periods with low plant abundances, mass seedlings, etc. should also be considered. Repeated measurements could also give insight for arid sites. For example, it currently remains unclear if the burrow entrance amounts in NP Pan de Azúcar are high because rain does not reset the yearly burrowing activity records, or the density of burrowing individuals is comparatively high, even if caused by fewer known species numbers.

A third consideration is the high variation around the positive correlations between diameter and tunnel lengths (Fig. 4). This may be an artefact of measurements limited to the first obstacle (see methods section). In addition, the excavation rate estimate is only a minimum. With measurements *including* the curvatures of the tunnels, the slopes of the correlation are expected to be steeper, possibly more similar to the regression shown for small animals. Identification of the burrow builders on species level and known burrow dimensions for the respective
species would also be an approach to improve the minimum estimation of excavation rates.

Finally, comprehensive datasets are valuable to improve the conceptual understanding of downhill sediment fluxes. To upscale the conclusions from the hillslope scale to catchment or regional scales, further parameters obtainable with methods from other disciplines would improve the results. Soil texture measurements, as it influences the amount of energy spent, and therefore the overall activity of the respective burrowing species (Price
and Podolsky, 1989) could provide improved insights into burrow distributions. Geomorphological techniques (see review by Viles, 2020) would provide additional information, such as the form of sediment traps beneath monitored areas that enable the estimation of total material moved downhill. Soil mixing rates and differences in mixing rates as dependent on the depth, and over time spans of decades to millennia, could be resolved by luminescence or cosmogenic nuclide techniques applied to depth profiles (Gray et al., 2020; Reimann et al., 2017;
Schaller et al., 2018). In addition, photogrammetry of mound modifications over time would be useful for estimating burrow related sediment fluxes at the surfaces from loose soil patches. UAV remote sensing techniques could serve to more rapidly estimate burrow density in areas without vegetation cover (Lawton et al., 2006). Lidar





data measuring vegetation structure could facilitate to identify the occurrence and abundance of vertebrate (Müller et al., 2010) and invertebrate assemblages (Müller et al., 2018; Vierling et al., 2011).

**5 Conclusions**

This review provides a synthesis of existing knowledge and new observations concerning the zoogeomorphic effect of burrowing animals in Chile. This is a first step of quantification of the complete burrowing animals' community at given sites on a hillslope scale and identified patterns that are some of the principal components that drive differences of animal burrowing effects on hillslopes along a climatic gradient. The quantification of

zoogeomorphic effects on hillslope scales is in its infancy and additional observations are needed. The key findings of this study are:

This study documents 45 burrowing vertebrate and 345 burrowing invertebrate species and species distribution summaries for both groups (Figs. 1D and 2). Most burrowing mammal species are present in the mediterranean climate of Chile. For invertebrates most known burrowing species are present in the semi-arid region of Coquimbo

(IV) and adjacent regions. A second invertebrate peak of high species diversity is in the humid-temperate area around the region of Biobío (VIII).

For the Chilean study areas, minimum excavation rates range between $0.34 \text{ m}^3 \text{ ha}^{-1} \text{ yr}^{-1}$ for the arid site, $0.56 \text{ m}^3 \text{ ha}^{-1} \text{ yr}^{-1}$ for the semi-arid site, $0.93 \text{ m}^3 \text{ ha}^{-1} \text{ yr}^{-1}$ for the mediterranean site and $0.09 \text{ m}^3 \text{ ha}^{-1} \text{ yr}^{-1}$ for the humid-temperate site, with the latter being vastly underestimated due to vegetation and litter cover. Relative to single

species rates from literature, our calculated excavation rates are in the range of previous invertebrate rates and low for rates of vertebrates.

In this study we discuss the relationships between burrowing animal activity and different metrics such as topographic slope and aspect, and latitudinal site specific variations in solar radiation and climate. We found more burrow entrances on north-facing than on south-facing slopes. On the climate gradient animal burrow entrances

showed the highest activity in the semi-arid Santa Gracia study area (Fig. 3), and a gradual decrease in NP La Campana and lowest activity in NP Nahuelbuta – following decreasing MAT and increasing MAP as well as decreasing incoming solar energy towards the south. Arid NP Pan de Azúcar reflected less activity than the semi-arid region, potentially due to the extreme conditions that are more challenging for animals to adapt to.

**Author contribution**

KÜ and TAE conceived the study. KÜ designed the experiments and carried them out, with contribution of LP. KÜ, SB and JPA conducted the literature survey. KÜ prepared the manuscript and figures with contributions from TAE and JPA. All co-authors reviewed and improved the final manuscript.

**Competing interests**

The authors declare that they have no conflict of interest.

**Acknowledgements**

We acknowledge support from the German research foundation (DFG) priority research program SPP-1803 "EarthShape: Earth Surface Shaping by Biota" grant DFG EH329/17-1 and 17-2 to TAE. We thank the Chilean National Forestry Corporation (CONAF) for providing access to the sample locations and on-site support for the research. We are grateful for the help with solar radiation calculation to Jörg Bendix (Univ. Marburg, Germany),

species identification to Jaime R. Rau (ULAGOS, Chile), Guillermo D'Elía (UACH, Chile), for help with data acquisition we thank the CONAF staff of NP Pan de Azúcar. We thank the International Union for Conservation of Nature (IUCN) for providing the Chilean burrowing vertebrate distributions as shape files online (2018). We thank Nina Farwig for comments on an earlier version of the manuscript. We also acknowledge Andrés A. Ojanguren-Affilastro and Martín J. Ramírez (MACN, Argentina), Elizabeth Chiappa (UPLA, Chile) and Jose

Mondaca (SAG, Chile) for natural history of arthropods, and Arturo Cortes (ULS) and Juan P. Castillo (ULS) for natural history of *Callopistes*. JPA thanks DIDULS PR19231210 of the Universidad de La Serena, La Serena (Chile).

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
