# Peer review of "Reviews and Syntheses: Composition and Characteristics of Burrowing Animals along a Climate and Ecological Gradient, Chile"

_Biogeosciences, 2021_

## Referee Comment (RC2)

[referee-annotated manuscript omitted]

---

## Author Response (AR1)

**Point-by-point response to the reviews**

Dear editor,
dear reviewers,
thank you for your comments. We have adopted almost all the suggested comments.
For the item of referee 1 to include more photos please find the reasons for our decline below, including existence and copyright reasons. We agree with nearly all other comments and provided additional phrases or paragraphs for clarification where applicable and clarified details in figures and tables and respective captions. We hope that we fulfilled the referees expectations to their satisfaction. For two minor comments of referee #2 Simon Mudd no changes seemed applicable, the answers provided below explain our hesitation.

We will upload two versions in the resubmission: a version with all changes tracked and a version with all changes accepted.

If the editor approves the responses then we are pleased to have our manuscript published in BioGeoSciences.

Kind regards,

Kirstin Übernickel and Co-authors

**RC1**: 'Comment on bg-2021-75', Anonymous Referee #1, 07 Jun 2021

This paper combines literature review data with field data to provide a thorough examination of the role of burrowing animals in varying environmental situations in Chile. The paper is an excellent example of combining field data with a thorough literature review. I do wish the paper had some photographs of a range of different burrow forms and from invertebrates, reptiles, birds, and mammals. The final edit (i.e. technical corrections) only needs a minor bit of tweaking of the English. Overall, I compliment the authors for a well conceived and constructed paper that was very interesting and presents important zoogeomorphic data from an under-represented part of the world.

Dear referee #1,
Thank you for your time reading our manuscript. We appreciate your comment on our manuscript and your kind evaluation.
Regarding the photographs we would have liked to be able to provide such insight. We included example entrance photos on bare ground and with the tunnel entrance in litter, see Figure 3 d, e. Unfortunately, the sampling at the sites did not include identification of species origin to the specific burrows, also the 390 species identified via the literature compilation make this attempt rather impossible. Many species' images are available in the internet but have copyright conditions that prohibit us from being able to use them. The reuse of a compilation would require a very large effort to get copyright permission for the 390 species. Some of the species specific references in our accompanying data publication include photos of specific species burrows, unfortunately only in very few cases. We are afraid the knowledge of species specific burrows for recognition in the field is hardly known and even less published.

We made use of our chance to correct some minor changes in the sentence structure to facilitate comprehension.

**RC2**: 'Comment on bg-2021-75', Simon Mudd, 22 Jun 2021

Dear Simon Mudd,
Thank you very much for your time that you spent on our manuscript and your constructive comments. We answer your general comments and also the specific comments per line from your pdf file. All changes to the manuscript are visible in the track changes version that we uploaded as well.

Part 1: general comments:

This manuscript has a few different components. Firstly, it reports detailed field measurements from four sites (each having two plots) where burrow densities were systematically sampled. This sampling involved counting burrows and recording their diameters and minimum lengths. The second component is a compilation of all the burrowing species in Chile. And a third component is a compilation of global studies that measured excavation rates.
For the first component, I felt the methods section (which in this paper is called "Data Compilation") is incomplete: some of the methods are presented later in the paper (for example how the volumes were calculated) and there is some text about how burrow angles are used that contradicts an earlier statement about what data was collected (see my commented pdf for the details).

Thank you. We moved the details you indicated from *2.2.2 Data analysis* to *2.2.2 Data compilation*.

The compilation of Chilean burrowing species was interesting, although some discussion of the sampling bias in this dataset would be useful. It isn't clear to me if the data on species diversity is real or just a function of proximity to major universities.

Thank you. We added a paragraph in the discussion section *4.1 Literature compilation on Chilean burrowing animals* about biases in the literature dataset.

Then there is a data compilation on excavation rates. This is very interesting: which animals are most efficient at moving sediment? But it is a big challenge to actually quantify how much sediment is being moved by fauna.

We appreciate your comment. To emphasize the difficulty, we added phrases in the introduction and the discussion section *4.2.6 Minimum excavation rates.*

In this manuscript the volumes for the burrows appear to be calculated using the minimum length of the burrow multiplied by the cross sectional area of the burrow (although this could be clearer in the methods).

Thank you. We moved the explanation of the calculation from section *2.2.3 Data analysis* to *2.2.2 Data compilation*.

It isn't clear how this is converted into a flux. Is it assumed the burrows are newly dug each year? Presumably the other papers listed in the compilation have some mixture of methods to estimate both volumes and fluxes. This isn't listed in the paper. But I think it should be so that readers know if some of the methods are more likely to be underestimates and which ones have more carefully calculated fluxes. I am most familiar with the Gabet 2000 paper, mainly because the author of that paper and I were graduate students together at the time and the author spent many, many hours complaining about how difficult it was to excavate complete gopher burrows. That paper also got volumes from terminal sediment piles. So I have high confidence in that data. Are other fluxes as reliable? I would like to have that information. It seems the method of flux estimation from each study could be added to the compilation table as a new column.

Thank you. We clarified regarding our own method in section *2.2.2 Data compilation*.
Regarding the compilation of published excavation rates, we added the requested column of the method applied for rate calculation in table 1 and stress this addition in the discussion section *4.1.3. Burrowing animal excavation rates*.

Finally, I think that a stronger connection between the survey of burrowing species and the excavation rates could be made. The species densities are difficult to quantify (and for most species this data is unavailable) but it would be interesting to discuss which species might be dominant in moving sediment from bioturbation in the different regions of Chile. I suppose I do not have a specific revision request here but more of a feeling that the survey of burrowing species feels somewhat disconnected from the site-specific work where there was an emphasis was quantifying fluxes.

> Thank you. We added a paragraph in section discussion *4.1.1 Relationship between burrows and climate* regarding the literature survey. This is the best we can do with available information in the literature.

The site work meshes quite well with the global survey of excavation rates, whereas the species survey seems bolted on. I leave it up to the authors to decide how best to do this.

> Thank you. We added a paragraph to the introduction explaining our dual approach.

Part 2: Comments in pdf file:

l. 41 "Why is this word and zoomgemorphologic in quotation marks? I don't think the quotation marks are needed."

> Thank you. We erased the quotation marks.

l. 50-52 "Awkward sentence. Suggest rewriting."

> Thank you. We rephrased the sentence.

l.104 So species that use burrows by non sympatric species are included? Are you not just excluding any species that uses a burrow dug by another species?

> Thank you. We replaced „*sympatric*" by "*other*"

l. 105 Why were these excluded?

> We extended the respective paragraph *2.1 Literature compilation: Burrowing animals in Chile* providing the reason.

l. 121 / Figure 1 What are the stars? Not in the caption. Say they are the field sites here.

> Thank you. Included.

l. 121 / Figure 1 What are the black areas in these plots?

> Thank you. Included.

l. 129-131 This needs more explanation. Presumably the roman numerals denotes the region: how are these demarcated and here do they come from?

> Thank you. We included the requested information in the caption.

l. 144 Say that the colour is from the elevations.

> The figure contains legends to the coloring of every map shown, including the information that a) contains shades of green for elevation.

l. 180-181 Presumably there is a minimum size of burrow you could detect? Can you get ant mounds? I think some comment about the scale of observable burrows would be relevant here.

> Thank you. We added the minimum diameter of burrow entrances measured, 1 mm, in the methods section *2.2.2 data compilation*.

l. 185-186 Was any excavation attempted to see if the minimum tunnel lengths had any relationship at all to the length of the tunnel? Or were volumes in the terminal mound measured

to see if the minimum tunnel length could account for a particular fraction of the volume in the terminal mound?

> We did not attempt any cross-validation of our measurements. Regarding comments above we added a passage in section *4.1.3 Burrowing animal excavation rates* including this information, although we do note the text does already mention that rates we observed in Chile are comparable to similar (sparse) rates available in the literature.

l. 186-187 Later the depth of burrows is reported. How is this calculated if only vertical, horizontal, or 45 degrees was recorded?

> Thank you. We moved the respective information from *2.2.3 Data analysis* to *2.2.2 Data compilation*.

l. 201 / Figure 4: What constitutes a large or a small animal? Say in the caption.

> Thank you. We added this in the caption.

l. 201-202 / Figure 4: These aren't tunnel lengths, they are minimum tunnel lengths. There could be a big difference! It needs to say on the axis label and in the caption these are minimum lengths.

> Thank you. We changed both to "minimum tunnel length" and also adopted the caption.

l. 214 The methods don't explain how depths were measured.

> Thank you. We moved the information from section *2.2.3 Data analysis* to *2.2.2 Data compilation*.

l. 221 The method (or data collection) section does not explain how volumes were calculated.

> Thank you. We moved this information from section *2.2.3 data analysis* to *2.2.2 data compilation*.

l. 229-230 Why is this not in the methods? And in the method it says only horizontal, vertical or 45 degrees were recorded.

> Yes, we moved this information from *2.2.3 Data analysis* to *2.2.2 Data compilation*.

l. 231 So these are minimum tunnel volumes? The methods section is missing a lot of important details.

> Thank you. we moved the information about the minimum tunnel length from the section *2.2.3 Data analysis* to *2.2.2 Data compilation* and included "minimum".

l. 233 Put this in the methods.

> Thank you. we moved this phrase from *2.2.3 Data analysis* to *2.2.2 Data compilation*.

l. 236 The ordering of the text needs some work (see previous comments).

> We hope we have accomplished this comment with the adaptations we made to the text regarding previous comments.

l. 318 Mixing of common and latin names here is a little confusing. So Ctenomys is a tuco-tuco? In the sentences below the common names are mixed with taxonomic names, maybe this could be done here? Note I am no biologist so ignore these comments if you are following convention.

> Thank you. We added common names for the species of *Ctenomys* and *Chelemys* mentioned in this paragraph.

l. 329 Smallest amongst what? Surely ants have smaller entrances. Is this amongst mammals? Or rodents?

> Thank you. we modified the text in *3.1.1 Burrowing vertebrates* adding "among mammals"

l. 342 It there any way to estimate sampling bias? That is, do you know if southern Chile has fewer species because there are fewer, or because it is farther away from the main universities? It is wet and not that cold, so one might expect there to be plenty of burrowing species in the south (same question applies to section on mammals).

*Thank you. We added a paragraph regarding this topic. See also the respective general comment above.*

l. 417 Very interesting compilation.

*Thank you.*

l. 476 This should be on a per area basis.

*We are not sure if we understood your comment here. The analysis of number of entrances was done across slopes within each site (Figure 8) and also along the climate gradient (Figure 10). To stress this point we added "per study site" in the commented phrase. We consider both analysis of value.*

l. 493 / Figure 10 I think tis should be "number" instead of "amount"

*Thank you. Changed.*

l. 511-512 How was the annual rate (or a volume per time) computed? Are you assuming the entire minimum volume estimated from the dimensions of the burrow is excavated each year? More discussion of this component needs to be in the methods section.

*Thank you, we moved this information from 2.2.3 Data analysis to 2.2.2 Data compilation*

l. 543-545 Some discussion about the way these numbers were estimated is warranted. How confident are we in the numbers? Are these volumes estmated from burrow sizes and the rate is assumed? That is, is an annual turnover of burrows assumed?

*Thank you. We moved this information from 2.2.3 Data analysis to 2.2.2 Data compilation. We added explanation in 4.1.3 Burrowing animal excavation rate that also addresses this comment.*

l. 552-553 This paper needs a more detailed discussion on the flux rates that can be informed by these previous papers.

*Thank you. We rephrased section 4.1.3 Burrowing animal excavation rates including a more detailed discussion.*

l. 627-628 This seems very speculative (you don't have sites with similar MAT/MAP but different ENSO intensity) so this discussion might be better alluded to in the conclusions.

*Agreed. We deleted the paragraph about ENSO effects and the respective references.*

l. 647 Again, how do you know the volumes measured were not from old burrows? How did you calculate a rate?

*Thank you. We moved this information from 2.2.3 Data analysis to 2.2.2 Data compilation.*

---

## Author Response (AR2)

**Response to the minor revision**

Dear editor Steven Bouillon,
dear referee #2 Simon Mudd,
thank you for your comments. We have adopted all the suggested comments.

We will upload two versions in the resubmission: a version with all changes tracked and a version with all changes accepted.

If the editor approves the responses then we are pleased to have our manuscript published in BioGeoSciences.

Kind regards,

Kirstin Übernickel and Co-authors

**Referee #2**: Simon Mudd, 05 Aug 2021

minor revisions:
I have now read the authors' responses to my queries, and am satisfied with their answers. I do think the lack of full excavation might substantially underestimate sediment fluxes, but the authors make this quite clear in the text and it is also clear that full excavation was not an option at their field sites. The more speculative passages have been removed. And I think the data compilation will be very useful to future workers in this area. I look forward to seeing this paper in print in the future. I have only minor recommendations for changes:

> Dear Simon Mudd,
> Thank you very much for your time that you spent on our manuscript, the positive evaluation of our work and the revision remarks. We have adopted all suggested comments. All changes to the manuscript are visible in the track changes version that we upload as well.

Line 50: Is the "of composition" necessary in this sentence?
> Removed.

Line 65: Remove comma after "impossible"
> (now l. 57) Removed.

Line 111: Delete comma after list and replace "credits" with "includes"
> (now l. 95) Deleted and replaced.

Line 115: Another use of the "composition". I'm not quite sure what you are trying to convey. Would "the effect of local burrowing species" not work as well? If the composition part is really meaningful can you include an extra sentence to explain this?
> (now l. 99) Thank you. We removed "composition" where the expression referred to the local burrowing species.

Line 214: I suggest: "The ecosystem is disturbed by (illegally) grazing cows." (a less passive sentence)
> (now l. 191) Changed.

Line 639: Clunky sentence. Maybe "Our data compilation was complicated by the fact that the primary aim of most of the included studies was not to quantify rates of sediment transport by burrowing." (or something like that).

(now l. 560) Adapted.

Line 641: delete "especially". Not needed.

(now l. 561) Deleted.

Line 689: Suggest "create mounds" as opposed to "do mounding"

(now l. 607) Changed.

Line 691: "is a minimally invasive…"

(now l. 610) Changed.